# Symmetry-Guaranteed Prediction of High-Order Tensor Properties for Crystalline Materials via Irreducible Decomposition

## Abstract

Predicting high-order tensor properties for crystalline materials is crucial for various scientific and engineering applications. Crystal symmetry is one of the primary factors influencing high-order tensor properties, such as elasticity and piezoelectricity, making strict adherence to symmetry constraints essential. However, exactly guaranteeing symmetry compliance remains challenging. Recent approaches rely on enforcing symmetry but often fail to strictly preserve symmetry. In this work, we propose a novel method that guarantees exact symmetry compliance by predicting symmetry-constrained irreducible components of high-order tensors. Specifically, we first develop a computational procedure to identify the basis tensors corresponding to symmetry-constrained irreducible components under various symmetry conditions. This symmetry-constrained basis guarantees that the assembled full tensor strictly adheres to the required symmetry constraints. To predict the numerical values for these irreducible components, we then propose a spherical-harmonic convolutional neural network designed to effectively capture essential high-order tensor information. Extensive experiments validate that our method achieves exact symmetry compliance without compromising prediction accuracy, thereby outperforming state-of-the-art approaches.

## 1 Introduction

Tensorial properties of crystalline materials, such as dielectric permittivity, piezoelectric coefficients, elastic stiffness, and higher-order nonlinear optical susceptibilities, are fundamental for technological advancements in electronics, photonics, energy harvesting, and quantum optics (Lovett, 2018; Petousis et al., 2016; Xiao et al., 2020). The accurate prediction of these anisotropic and orientation-dependent properties directly impacts material discovery and design, as their precise characterization is intimately connected to the performance of crystalline materials in practical applications. Achieving reliable predictions requires careful consideration of the intrinsic symmetry properties inherent in crystalline structures.

Indeed, crystal symmetry significantly governs tensor properties by constraining how tensors transform under symmetry operations, thereby determining a material's macroscopic behavior. For instance, materials with cubic symmetry, such as diamond, exhibit uniform deformation under stress, whereas those with hexagonal symmetry, such as graphite, display anisotropic deformation behaviors dependent on crystallographic directions (Noya et al., 2010; Jain et al., 2014; Gray et al., 2009). Moreover, even within a nominal symmetry class, slight deviations or symmetry-breaking distortions can induce profound changes in response; for example, the cubic-to-tetragonal distortion in perovskites such as $BaTiO_3$ gives rise to ferroelectricity and markedly different elastic and dielectric behaviors (Jia et al., 2025; Hashemizadeh et al., 2016). Therefore, explicitly incorporating symmetry constraints into predictions is essential for accurately modeling tensorial behavior; yet strict enforcement, particularly for higher-order tensors such as the elastic tensor, remains computationally and methodologically challenging.

Although first-principles methods, most notably Density Functional Theory (DFT), yield quantitatively reliable and symmetry-consistent predictions across a broad range of properties, their poor scaling and the need for large supercells and dense Brillouin-zone sampling render

them prohibitively expensive for large systems, complex configurational spaces, and high-throughput studies (Kohn & Sham, 1965; Kohn et al., 1996). To bypass this bottleneck, machine learning (ML) surrogates are increasingly used to predict tensorial properties at a fraction of the cost; however, the speed gains often come at the expense of fidelity. In practice, ML models provide only approximate emulations of DFT and, crucially, struggle to rigorously preserve the directional dependence and crystallographic symmetry constraints that tensors must satisfy. Many widely adopted frameworks were originally developed with scalar targets in mind—via descriptor-based methods (Dunn et al., 2020; Kỳvala et al., 2025; Himanen et al., 2020; He et al., 2021) or (rotation-)invariant graph neural networks (Chen et al., 2019; Xie & Grossman, 2018; Chen & Ong, 2022; Kang et al., 2023; Fan et al., 2022; Yan et al., 2022; Choudhary & DeCost, 2021), and when repurposed for tensors they would violate index-level symmetries, equivariance, and crystal point-group constraints, leading to biased or internally inconsistent predictions that become more pronounced for higher-order tensors (e.g., elasticity or piezoelectricity).

To address these shortcomings, recent developments in equivariant neural networks aim to explicitly enforce symmetry transformations on predicted tensors (Schütt et al., 2021; Batzner et al., 2022b; Batatia et al., 2022; Yan et al., 2024b; Grega et al., 2024; Xu et al., 2021; Van't Sant et al., 2023). Despite notable improvements, such symmetry enforcement approaches remain approximate, relying predominantly on crystal-specific rotation or reflection matrices to incorporate symmetry constraints. While effective for lower-order tensors, these methods become increasingly problematic as tensor complexity and order rise, leading to subtle but significant symmetry violations (Grisafi et al., 2018; Wen et al., 2024; Jekel et al., 2022). These discrepancies become critical for properties where strict symmetry compliance is mandatory. For example, elastic tensors (4-th order), which determine properties like Young's modulus, require exact symmetry adherence to ensure physical consistency (e.g., positive definiteness) Grega et al. (2024); Kalra et al. (2016). Even minor symmetry violations in high-order tensor predictions can drastically impair the accuracy of subsequent calculations and predictions, undermining their reliability and utility for practical engineering applications.

In this work, we propose an exact, symmetry-guaranteed framework for predicting high-order tensor properties of crystalline materials by learning in the irreducible-component space rather than in the full-tensor space. The core guarantee is that symmetry is satisfied regardless of the prediction accuracy, since even if the learned magnitudes of the irreducible components are imperfect, the reconstructed target tensor obeys the crystallographic constraints by construction. This follows from three facts: (i) any high-order tensor subject to a crystallographic symmetry admits an exact decomposition into a finite set of irreducible components, (ii) a unique reconstruction of the target tensor is obtained via a combination of these components, and (iii) symmetry enforcement is straightforward for the irreducible components via canonical bases or projection, in contrast to the fragile high-order constraints required on the full tensor. Building on this pipeline, we enumerate all the appropriate irreducible components and reconstruct tensors exclusively from them, so any output is symmetry compliant by design. We then use a spherical-harmonic convolutional neural network to predict the magnitudes of irreducible components. Moreover, since the target tensors are high-dimensional and difficult to predict as a whole, decomposing them into irreducible components with straightforward symmetry guarantees has advantages in achieving better accuracy. Specifically, by reconstructing the target tensor using the irreducible components, the search space has been reduced to a small number of degrees of freedom, resulting in accuracy improvement while preserving exact symmetry.

## 2 RELATED WORK

**Datasets** Datasets of tensor-valued material properties (e.g., elasticity and piezoelectric tensors) are critical for modeling complex anisotropic behavior and are increasingly produced by first-principles calculations such as DFT (Kohn et al., 1996). Prominent public repositories like the Materials Project (Jain et al., 2013) and JARVIS-DFT (Choudhary et al., 2020) now curate extensive collections of these computed tensor properties, facilitating large-scale data-driven materials discovery. These datasets typically include not only high-order tensor properties but also other essential material information such as atomic positions, edge

connectivity (bond order), and crystal constants. Additionally, they encompass related downstream properties, such as Young's modulus, shear modulus, and other mechanical, thermal, or electronic properties, which are vital for evaluating the overall performance and behavior of materials under various conditions.

**Equivariant message passing network**   Equivariant Message Passing Neural Networks (MPNNs) (Batatia et al., 2022; Anderson et al., 2019; Batzner et al., 2022a; Deng et al., 2023; Satorras et al., 2021; Schütt et al., 2021) are a type of Graph Neural Networks (GNNs) designed to efficiently handle Euclidean symmetry operations, such as rotations, reflections, and translations. In most MPNNs, messages are propagated using a set of spherical bases, allowing the model to learn not only vector-based features but also high-order tensor features across different layers. Thus, MPNNs become a powerful tool for learning the representations of crystalline materials.

**Methods for enforcing symmetry in high-order tensor properties**   To enforce symmetry in high-order tensor properties, many models adopt different strategies, such as incorporating space group symmetries (Yan et al., 2024b) or using matrix decomposition (Xu et al., 2021; Van't Sant et al., 2023) techniques. While these methods help in aligning the learned tensor representations with the expected symmetries, most of them rely on approximate symmetry enforcement, rather than ensuring exact symmetry compliance.

## 3 Methods

This section is devoted to the prediction of high-order tensor properties in crystalline materials and is structured into three complementary components: **theory, computational procedure and model design**. (1) We begin with a rigorous group-theoretic analysis in Section 3.1 demonstrating that any rank-$k$ tensor defined under the rotation group SO(3), even with external symmetry constraints, admits a finite and exact decomposition into irreducible subspaces. This lays the mathematical groundwork for subsequent algorithmic and architectural developments. By assembling these basis tensors back into the full tensor, one inherently guarantees exact symmetry compliance, which lays the mathematical groundwork for all subsequent algorithmic and architectural developments. (2) Building upon this theoretical result, we develop an efficient computational procedure in Section 3.2 to determine the set of irreducible representations that remain valid under specific symmetry constraints. This enables a systematic identification of relevant subspaces for accurate tensor prediction. (3) Finally, in Section 3.3, we introduce an equivariant neural network with spherical-harmonic convolutions that operates directly in irreducible spaces to predict high-order tensor components. This design ensures compliance with SO(3)-equivariance and enables both physical consistency and predictive accuracy.

### 3.1 Theory for high-order tensor decomposition

An essential step in predicting high-order tensor properties is to decompose the tensor into its irreducible components and represent it as a direct sum of these fundamental parts under the action of the SO(3) rotation group. In this section, we demonstrate how such a decomposition can be efficiently performed using group-theoretic principles.

We begin by presenting two propositions to illustrate the logic behind the irreducible decomposition of high-order tensors:

**Proposition 1**   Let $V$ be a three-dimensional Euclidean vector space. Any rank-$k$ ($k > 1$) tensor $\mathcal{T}_{i_1 \cdots i_k} \in V^{\otimes k}$ admits a unique decomposition into irreducible representations (irreps) of SO(3) group labeled by integer spins $\ell$ as follows:

$$V^{\otimes k} \simeq \bigoplus_{\ell \in L_k} \left( \mathcal{S}_k^{(\ell)} \bigoplus \mathcal{A}_k^{(\ell)} \bigoplus \mathcal{M}_k^{(\ell)} \right), \tag{1}$$

where $L_k$ is the set of all distinct integer spins $\ell$ that appear in the decomposition of $V^{\otimes k}$. $\mathcal{S}_k^{(\ell)}$, $\mathcal{A}_k^{(\ell)}$, and $\mathcal{M}_k^{(\ell)}$ denote the symmetric, the antisymmetric, and the mixed-symmetry components transforming under spin-$\ell$ irreps, respectively.

**Proposition 2** Let $\mathcal{T}$ be a rank-$k$ tensor defined over a three-dimensional Euclidean space. Under decomposition into irreducible representations (irreps) of SO(3) group: When $\mathcal{T}$ satisfies additional index symmetries (e.g., major/minor symmetry, tracelessness, or antisymmetry), the set of allowed irreps forms a proper subset of the unconstrained case: $\mathrm{Irrep}(\mathcal{T}_{\mathrm{sym}}) \subsetneq \mathrm{Irrep}(\mathcal{T}_{\mathrm{unconstrained}})$.

Together, Propositions 1 and 2 (whose proofs are detailed in the Appendix A.1 establish a clear roadmap for exact, symmetry-aware tensor decomposition under SO(3): **Unconstrained decomposition**. Any rank-$k$ tensor can be uniquely split into a finite direct sum of irreducible subspaces—symmetric traceless, antisymmetric, and mixed-symmetry components—each carrying a well-defined spin $\ell$ and dimension $2\ell + 1$. **Effect of additional symmetries**. Imposing index symmetries (e.g. major/minor index exchange, tracelessness, antisymmetry) simply eliminates those irreducible representations that are incompatible with the constraint. In other words, the set of allowed irreps under symmetry is a strict subset of the full unconstrained set.

These results imply that once we have computed the full SO(3)-irreducible decomposition of a tensor via Proposition 1, we obtain its symmetry-constrained form by discarding the forbidden irreps identified in Proposition 2. Specifically, by assembling the components based on their basis, we ensure that the resulting full tensor satisfies all symmetry constraints. This process guarantees that the final decomposition is exact and finite, inherently respecting all imposed symmetries, thus providing a solid theoretical foundation for our prediction model based on irreducible components.

### 3.2 COMPUTATIONAL PROCEDURE FOR COMPUTING IRREDUCIBLE DECOMPOSITION

Algorithm 1 implements a two-step method to extract all symmetry-compatible spin-$\ell$ components of a rank-$k$ tensor under an arbitrary point-group symmetry $G_{\mathrm{sym}}$:

**Unconstrained Decomposition** Based on **Proposition 1**, we introduce the procedure GENERATESO3IRREPS, which enumerates all symmetric components $\ell = k,\ k-2,\ k-4,\ldots$, all antisymmetric components $\ell = 2, 3$ and mixed-symmetric components to record the multiplicity of each $\ell$ in a temporal map $\mathcal{M}_0[\ell]$ Brachat et al. (2010); Gusev et al. (2015); Costa & Hansen (2015). (see more details in Appendix A, B and F)

**Symmetry Compatibility** Starting from the unconstrained decomposition $\mathcal{M}_0$, the function APPLYSYM maintains only the degrees of freedom that are invariant under symmetry constraints $G_{\mathrm{sym}}$. Specifically, for each component $(\ell, m_\ell)$ in $\mathcal{M}_0$, we first calculate a class sum over discrete symmetry operations in $G_{\mathrm{sym}}$: $S_\ell = \sum_{C_j \subseteq G_{\mathrm{sym}}} |C_j| \, \chi_{O(3)}^{(\ell)}(\alpha_j, \det_j)$, where $\chi_{O(3)}^{(\ell)}(\alpha, \det) = s(\ell, \det)\, \chi^{(\ell)}(\alpha)$ is the character of the $\ell$-representation evaluated at a representative $(\alpha, \det)$ of each conjugacy class and $s(\ell, \det)$ is the parity correction factor that accounts for improper operations (Klebanov & Tarnopolsky, 2017; Higuchi, 1987). The invariant multiplicity is then calculated by $n_{\mathrm{triv}}^{(\ell)} = \max\{0,\ \mathrm{round}(S_\ell/|G_{\mathrm{sym}}|)\}$ (Tung, 1985; Lyubarskii, 2013). Consequently, we only record the usable coefficients as $\mathcal{M}_{\mathrm{final}}[\ell] = m_\ell \, n_{\mathrm{triv}}^{(\ell)}$ when $n_{\mathrm{triv}}^{(\ell)} > 0$. When an explicit basis is needed, we project the invariant subspace via $\Pi_\ell = |G_{\mathrm{sym}}|^{-1} \sum_{g \in G_{\mathrm{sym}}} \rho^{(\ell)}(g)$. In brief, APPLYSYM counts, for each $\ell$, the dimensions that remain unchanged under the symmetry constraint and keeps only those for reconstruction, guaranteeing symmetry by design.

### 3.3 MODEL DESIGN

We observe that the irreducible decomposition of high-order tensors aligns naturally with spherical harmonics, as both share the same SO(3) representation structure. Each irreducible component corresponds to a spherical harmonic with angular quantum number $\ell$, making spherical harmonics an ideal basis for modeling rotation-equivariant tensor properties.

Motivated by this insight, we specifically design our crystal tensor prediction model IrredNet (Figure 3.3) to incorporate spherical harmonic bases, enabling the direct construction of irreducible representations with built-in SO(3)-equivariance. This design not only respects

**Algorithm 1** SO(3) Irreducible Decomposition of Rank-$k$ Tensor under Symmetry Constraints $G_{sym}$

---

**Input:** Tensor rank $k$; Point group symmetry $G_{sym}$
**Output:** Final map $\mathcal{M}_{final}$ from irreducible decomposition of k-th order tensor

---

1: **function** CHARO3($\ell, \alpha, \det$)
2:    $\alpha \leftarrow \mod(\alpha, 2\pi)$
3:    $s \leftarrow \begin{cases} 1, & \det = +1 \\ (-1)^{\ell}, & \det = -1 \end{cases}$
4:    **if** $\alpha = 0$ or $\alpha = 2\pi$ **then**
5:        **return** $s \cdot (2\ell + 1)$
6:    **else**
7:        **return** $s \cdot \dfrac{\sin\big((\ell + 0.5)\alpha\big)}{\sin(\alpha/2)}$
8:    **end if**
9: **end function**

---

10: **function** ISCOMPAT($\ell, G_{\text{sym}}$)
11:    $S \leftarrow 0$
12:    **for** each $(|C_j|, \alpha_j, \det_j) \in G_{\text{sym}}$ **do**
13:        $\chi_j \leftarrow$ CHARO3($\ell, \alpha_j, \det_j$)
14:        $S \leftarrow S + |C_j| \cdot \chi_j$
15:    **end for**
16:    $n_\ell \leftarrow \left\lfloor \dfrac{S}{|G_{\text{sym}}|} + 0.5 \right\rfloor$
17:    **return** $n_\ell$
18: **end function**

---

19: **function** APPLYSYM($\mathcal{M}_0, G_{\text{sym}}$)
20:    $\mathcal{M}_{\text{final}} \leftarrow \emptyset$
21:    **for** each $(\ell, m) \in \mathcal{M}_0$ **do**
22:        $n_\ell \leftarrow$ ISCOMPAT($\ell, G_{\text{sym}}$)
23:        **if** $n_\ell > 0$ **then** $\mathcal{M}_{\text{final}}[\ell] \leftarrow \mathcal{M}_{\text{final}}[\ell] + m \times n_\ell$
24:        **end if**
25:    **end for**
26:    **return** $\mathcal{M}_{\text{final}}$
27: **end function**

**function** GENERATESO3IRREPS($k$)
    $\mathcal{M}_{\text{final}} \leftarrow \{\ell \mapsto 0\}$
  ▷ **Step 1:** Enumerate all partitions of $k$ with at most 3 rows
  Parts $\leftarrow$ GENERATEPARTITIONS($k, 3$)
    **for** each $\lambda = (\lambda_1, \lambda_2, \lambda_3) \in$ Parts **do**
  ▷ **Step 2:** Calculate multiplicity via Hook-length
        $\text{mult}_{\text{GL3}} \leftarrow$ HOOKLENGTHFORMULA($\lambda$)
    ▷ **Step 3:** Apply the SU(3) $\supset$ SO(3) branching
        $p \leftarrow \lambda_1 - \lambda_2$,
        $q \leftarrow \lambda_2 - \lambda_3$,
        $\mathcal{M}_{\text{branch}} \leftarrow$ BRANCHSU3TOSO3($p, q$)
    ▷ **Step 4:** Accumulate the results weighted by GL(3)
        **for** each $(\ell, m_\ell) \in \mathcal{M}_{\text{branch}}$ **do**
            $\mathcal{M}_{\text{final}}[\ell] \leftarrow \mathcal{M}_{\text{final}}[\ell] + m_\ell \times \text{mult}_{\text{GL3}}$
        **end for**
    **end for**
    **return** $\mathcal{M}_{\text{final}}$
**end function**

---

**MAIN**:
$\mathcal{M}_0 \leftarrow$ GENERATESO3IRREPS($k$)
**Verify** $\sum_\ell (2\ell + 1) \cdot \mathcal{M}_0[\ell] = 3^k$
$\mathcal{M}_{final} \leftarrow$ APPLYSYM($\mathcal{M}_0, G_{sym}$)
**return** $\mathcal{M}_{final}$

---

Additional functions GeneratePartitions, HookLengthFormula, BranchSU3toSO3 are described in detail in the Appendix F.

the intrinsic geometric symmetries of tensorial data but also facilitates more physically consistent learning, leading to better generalization and interpretability. It comprises three core components: (1) Crystal Graph Construction Block: building a graph based on the crystal structure; (2) Equivariant Interaction Block: propagating and aggregating features in an SO(3)-equivariant manner via a learnable gating mechanism; (3) Irreducible Decomposition Block: extracting and organizing the tensor's SO(3) irreducible components for final property estimation.

**Crystal Graph Construction Block** A crystal structure is defined by a $3 \times 3$ lattice matrix that encodes the three-dimensional periodicity of the unit cell, together with the set of basis atoms occupying that cell. To be specific, the structure can be mathematically represented by $\mathbf{M} = (\mathbf{A}, \mathbf{P}, \mathbf{L})$, where $\mathbf{A} = [\boldsymbol{a}_1, \boldsymbol{a}_2, \cdots, \boldsymbol{a}_n] \in \mathbb{R}^{d_a \times n}$ contains the $d_a$-dimensional feature vectors for $\mathbf{n}$ atoms in the unit cell, $\mathbf{P} = [\boldsymbol{p}_1, \boldsymbol{p}_2, \cdots, \boldsymbol{p}_n] \in \mathbb{R}^{3 \times n}$, $\boldsymbol{L} = (\boldsymbol{\ell}_1, \boldsymbol{\ell}_2, \boldsymbol{\ell}_3) \in \mathbb{R}^{3 \times 3}$ describes the repeating patterns of the unit cell structure in 3D space. To enable networks to handle such infinite crystal structures, we follow the strategy in previous work (Yan et al., 2024a; Wen et al., 2024; Batatia et al., 2022) to build a graph to describe structures and relationships. Each node in the proposed SE(3) invariant crystal graph represents the atom $i$ and all its infinite duplicates in the 3D space with positions $\{\hat{p}_i \mid \hat{p}_i = p_i + k_1\ell_1 + k_2\ell_2 + k_3\ell_3, \quad k_1, k_2, k_3 \in \mathbb{Z}\}$ and atom feature $a_i$. The edge is built from the source node $s$ to the target node $t$ when the euclidean distance $\|e_{ts}\|_2$ between a

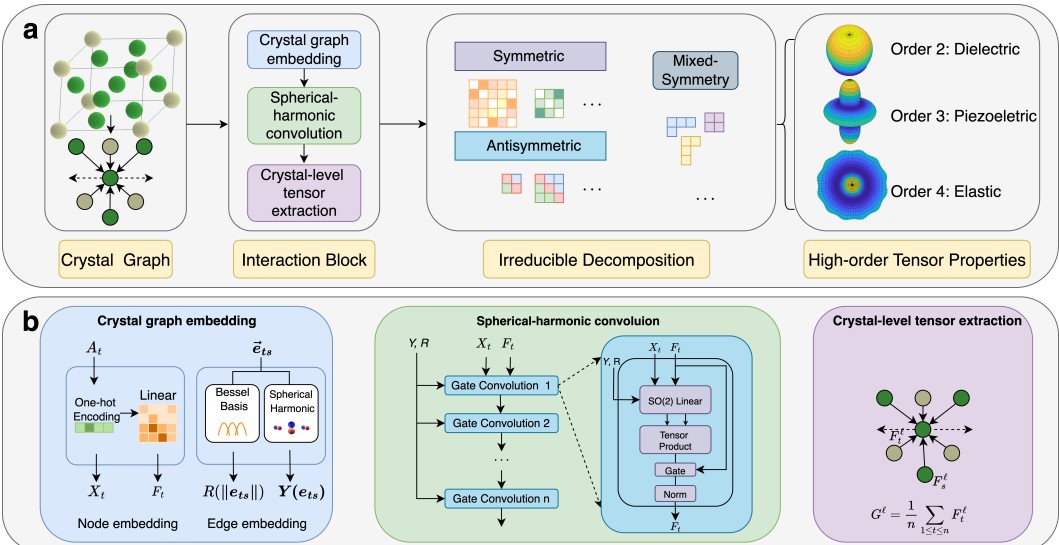

Figure 1: (a) Overview of IrredNet. A periodic crystal structure is first converted into a crystal graph and processed by several Equivariant Interaction Blocks to obtain SO(3)-equivariant node features. These features are then fed into the Irreducible Decomposition Block, where they are expanded on a spherical-harmonic basis, decomposed into irreducible components indexed by angular momentum $\ell$, and filtered according to the target tensor symmetries. Different colors indicate groups of spherical harmonics forming bases of irreducible components associated with different parts (dimensionalities) of the high-order tensor property. The remaining symmetry-adapted components are finally combined by the tensor head to reconstruct the physical tensor. (b) Internal structure of a single Equivariant Interaction Block used in (a). It consists of a crystal graph embedding module (initializing node and edge features from atomic species and interatomic geometry), a spherical-harmonic convolution module that updates features in an SO(3)-equivariant manner, and a crystal-level tensor extraction module that aggregates atomic features into crystal-level representations passed to the irreducible decomposition stage in (a).

duplicate of $t$ and $s$ in a periodical system satisfies $\|e_{ts}\|_2 \leq r$, where $r \in R$ is the cutoff radius determined by the $k$-th nearest neighbor.

**Equivariant Interaction Block**  This block is divided into three parts: the first part uses an embedding layer to obtain the basic representation of the graph; the second part employs equivariant spherical-harmonic convolution to capture high-order features; and the final part aggregates hierarchical order information at the crystal graph level. Following previous work (Yan et al., 2024a; Wen et al., 2024; Batatia et al., 2022), we map node type into a one-hot feature representations. We embed each edge length $\|e_{ts}\|$ by expanding it onto a preset family of basis functions, producing a fixed-length vector that captures both the raw distance and its multi-scale variations. Edge directional vectors $e_{ts}$ are embedded by corresponding spherical harmonics $Y(e_{ts})$.

To efficiently extract capture high-order features (especially for $\ell > 0$), we employ a spherical-harmonic convolution module that operates through several equivariant message-passing layers. Specifically, we replace the traditional SO(3) convolutions with Equivariant Spherical Channel Network Passaro & Zitnick (2023); Liao et al. (2023). Instead of performing full SO(3) convolutions, this framework reformulates the required tensor products as SO(2) linear operations, greatly reducing the computation cost. In a standard SO(3) convolution, one takes the irreducible representation features $\psi_{m_i}^{\ell_i}$ and spherical-harmonic projections $Y_{m_f}^{\ell_f}(\hat{e}_{ts})$ and computes their tensor product. By applying a rotation matrix $D_{ts}$ to align $\hat{r}_{ts}$ with the canonical axis, the rotated spherical harmonic $Y_{m_f}^{\ell_f}(D_{ts}\hat{r}_{ts})$, is nonzero only for the $\ell_f = 0$, $m_f = 0$ channel. This collapses the product to $C_{(\ell_i, m_i),(\ell_f, 0)}^{(\ell_o, m_o)}$, which itself is only

nonzero when $m_i = \pm m_o$. Leveraging this observation, eSCN replaces the original $O(L^6)$ SO(3) convolutions with $O(L^3)$ SO(2) operations on the rotated tensors (where L is the highest degree).

The spherical-harmonic convolution module stacks several gated convolution layers to efficiently propagate high-order feature representations. These gates filter out redundant scalar information, allowing the network to concentrate on more intricate, high-order interactions. Moreover, by applying an iterative training strategy at every layer, we further enhance the module's ability to learn and generalize complex, hierarchical features. Finally, after several equivariant message-passing layers are applied to extract high-order node features, the crystal-level features are aggregated from the node-level embeddings.

**Irreducible Decomposition Block** Based on the theory and computational procedure for the irreducible decomposition of high-order tensors under specific symmetry constraints, we can derive the basis for each irreducible component of the high-order tensor, as described before. A detailed implementation process for different high-order tensors is provided in Appendix.

## 4 EXPERIMENTS

### 4.1 EXPERIMENT SETTINGS

**Crystal high-order tensor dataset** In our study, we thoroughly evaluate our model IrredNet on the JARVIS-DFT dataset Choudhary et al. (2020), which includes crystal property tensors of three different orders. Following the preprocessing protocol established by previous work Yan et al. (2024b), we apply the same normalization procedure to ensure consistency. The dataset comprises dielectric, piezoelectric, and elastic tensors. We further summarize the physicochemical and geometric characteristics of these tensors in the following Table 1. Additionally, the dielectric property is represented by a $3 \times 3$ matrix $\boldsymbol{\epsilon}$ that satisfies the symmetry constraint $\boldsymbol{\epsilon}_{ij} = \boldsymbol{\epsilon}_{ji}$. The piezoelectric tensor $\boldsymbol{e}$ is a third-order tensor of size $3 \times 3 \times 3$, subject to the symmetry constraint $\boldsymbol{e}_{ijk} = \boldsymbol{e}_{ikj}$. The elastic tensor is a fourth-order tensor of dimension $3 \times 3 \times 3 \times 3$, governed by more complex symmetry conditions, including major symmetry ($C_{ijkl} = C_{klij}$) and minor symmetry ($C_{ijkl} = C_{jikl} = C_{ijlk}$). To provide a more compact and interpretable representation, the elastic tensor is often expressed using Voigt notation, which converts the tensor into a symmetric $6 \times 6$ matrix form containing 21 independent components. A detailed introduction is attached in the Appendix B.

**Baseline methods** We benchmark several state-of-the-art methods in the field, including MEGNet Chen et al. (2019), which is an invariant model commonly used for predicting tensorial data in previous work Morita et al. (2020). We also compare it to ETGNN Satorras et al. (2021), an equivariant model that excels at capturing tensorial features. Additionally, we evaluate GMTNet Yan et al. (2024b), another approach that uses space group-related rotational matrices to enforce symmetry and predict high-order tensor properties.

**Evaluation Metrics** To assess the quality of high-order crystal tensor predictions, we adopt several metrics from previous work Yan et al. (2024b); Hua et al. (2024); Petousis et al. (2016): (1) **Zero-element accuracy**: This metric evaluates the model's ability to accurately identify zero-valued entries in high-order tensors that arise due to symmetry constraints. (2) **Equality accuracy**: This metric assesses the model's capability to recognize mutually dependent tensor components that are constrained to have identical values due to intrinsic tensor symmetries. (3) **Frobenius norm (Fnorm) distance**: The Frobenius norm distance quantifies the discrepancy between the predicted and ground-truth tensors by computing the square root of the sum of the squared differences of all corresponding elements. It is defined as: $\|y_{\text{pred}} - y_{\text{true}}\|_F = \sqrt{\sum_i (y_{\text{pred},i} - y_{\text{true},i})^2}$. (4) **Error within threshold (EwT)**: EwT is used to evaluate the proportion of high-quality predictions that fall within specified relative error thresholds (e.g., 25%, 10%, 5%). It is defined based on the ratio between the Frobenius norm of the error and that of the ground-truth tensor: $\|y_{\text{pred}} - y_{\text{label}}\|_F / \|y_{\text{label}}\|_F$, where $y_{\text{pred}}$ and $y_{\text{label}}$ denote the predic ted and true tensor values, respectively. A prediction is considered high-quality if its EwT value falls below a specified threshold.

Table 1: Dataset statistics of high-order tensor properties.

| Dataset | # Samples | Fnorm | Unit | Order |
|---|---|---|---|---|
| dielectric | 4713 | $14.7\pm18.2$ | Unitless | 2 |
| piezoelectric | 4998 | $0.43\pm3.09$ | $C/m^2$ | 3 |
| elastic | 14200 | $327\pm249$ | GPa | 4 |

Table 2: Zero–element and equality accuracy of predicted rank-4 elastic tensors on the JARVIS-DFT test set, broken down by crystal system. For each crystal system, Zero reports the percentage of tensor components that are required to be zero by symmetry and are correctly predicted as zero, and Equality reports the percentage of pairs of components that are required to be equal and are predicted to satisfy this equality. Higher values indicate better symmetry compliance.

| Crystal System | MEGNet | | ETGNN | | GMTNet | | IrredNet | |
|---|---|---|---|---|---|---|---|---|
| | Zero | Equality | Zero | Equality | Zero | Equality | Zero | Equality |
| Cubic | 0% | 0% | 13.9% | 23.1% | 46.6% | **100%** | **100%** | **100%** |
| Tetragonal | 0% | 0% | 5.3% | 6.3% | 72.1% | 98.5% | **100%** | **100%** |
| Hexagonal | 0% | 0% | 2.5% | 7.2% | 59.4% | **100%** | **100%** | **100%** |
| Trigonal | 0% | 0% | 1.6% | 2.8% | 5.5% | 29.6% | **100%** | **100%** |
| Orthorhombic | 0% | 0% | 0% | 0% | 52.3% | **100%** | **100%** | **100%** |
| Monoclinic | 0% | 0% | 6.7% | 0% | 79.5% | **100%** | **100%** | **100%** |

Followed by previous works Yan et al. (2024b), we re-implement MEGNet, ETGNN, and GMTNet as our baselines, all with their default hyperparameters. Each dataset is randomly split into training, validation, and test subsets in an 80:10:10 ratio. We train our model using the Huber loss, the AdamW optimizer, and a polynomial decay schedule with an initial learning rate of $10^{-5}$. For the tensor datasets, we set the maximum spin quantum number $\ell$ to 3, 3, and 4 for the dielectric, piezoelectric, and elastic tensors, respectively. Detailed settings are listed in Appendix D and J.

### 4.2 Symmetry prediction of high-order tensor

To assess our model's ability to capture crystal symmetry across the crystal systems, we employ two accuracy metrics mentioned before: (1) **Zero-element accuracy** and (2) **Equality accuracy**. These two metrics assess (i) the proportion of tensor entries that are truly zero and correctly predicted as zero, and (ii) the proportion of nonzero entries whose predicted values satisfy the required equivariance constraints. Following the GMTNet setting, we evaluate symmetry compliance using two success-rate metrics: zero-element accuracy with a tolerance of $10^{-5}$ and equality accuracy with a tolerance of $10^{-4}$. Specifically, we assess symmetry preservation on the fourth-order elastic tensor, one of the most challenging cases in Table 2. Our model achieves perfect zero-element and equality accuracies and strictly preserving all symmetry constraints. In contrast, MEGNet and ETGNN fail to enforce these constraints, with zero-element accuracy near 0 % and equality accuracy only around 10–20 %. Although GMTNet which leverages space group information to enforce symmetry performs substantially better, its accuracy still degrades dramatically on more complex systems, such as trigonal crystals.

### 4.3 Accuracy of High-Order Tensor Predictions

We also benchmark our model's accuracy against several state-of-the-art baselines across three tensor datasets. Table 3 presents the prediction results of various models evaluated using the Fnorm metric and the EwT metrics at the 25 %, 10 %, and 5 % thresholds. Our model delivers three key advantages in high-order tensor prediction: it consistently reduces reconstruction error (Fnorm) from 3.50 to 3.23 on dielectric, 0.37 to 0.29 on piezoelectric, and 68.12 to 64.23 on the elastic tensor; it achieves the highest EwT scores even at the strict 5 % threshold: 29.8 % (dielectric), 49.2 % (piezoelectric) and 12.1 % (elastic), demonstrating superior capture of the fine-grained, symmetry-driven structure; and its gains grow with tensor complexity, peaking on the challenging fourth-order elastic tensor, underscoring the power of our high-order equivariant design.

Table 3: Comparison of high-order tensor prediction on the JARVIS-DFT dielectric, piezo-electric (Piezo), and elastic benchmarks. The table reports results for MEGNet, ETGNN, GMTNet, and the proposed IrredNet. For each method and dataset we show the Fnorm error ($\downarrow$) of the predicted tensors and EwT metrics ($\uparrow$), which give the percentage of test samples whose relative Fnorm error is below 25%, 10%, and 5%, respectively. The last row (Total Time (s) ($\downarrow$)) reports the total wall-clock time in seconds for training and evaluation on the dataset under identical hardware and batch-size settings.

| Metric | MEGNet | | | ETGNN | | | GMTNet | | | IrredNet | | |
|---|---|---|---|---|---|---|---|---|---|---|---|---|
| | Dieletric | Piezo | Elastic | Dieletric | Piezo | Elastic | Dieletric | Piezo | Elastic | Dieletric | Piezo | Elastic |
| Fnorm ($\downarrow$) | 4.16 | 0.49 | 132.21 | 3.92 | 0.40 | 90.84 | 3.50 | 0.37 | 68.12 | **3.23** | **0.29** | **64.23** |
| EwT 25% ($\uparrow$) | 74.9% | 46.1% | 22.8% | 81.3% | 47.1% | 42.0% | 84.5% | 49.1% | 65.0% | **85.8%** | **49.3%** | **66.5%** |
| EwT 10% ($\uparrow$) | 38.9% | 44.3% | 3.0% | 41.6% | 46.9% | 12.8% | **57.1%** | 46.3% | 21.6% | 53.2% | **49.3%** | **26.5%** |
| EwT 5% ($\uparrow$) | 19.1% | 41.1% | 0.3% | 23.8% | 46.8% | 1.2% | 27.8% | 45.7 % | 7.7% | **29.8%** | **49.2%** | **12.1%** |
| Total Time (s) ($\downarrow$) | 1005 | 381 | 14666 | 1510 | 447 | 34157 | 881 | **286** | >67000 | 798 | 288 | **14121** |

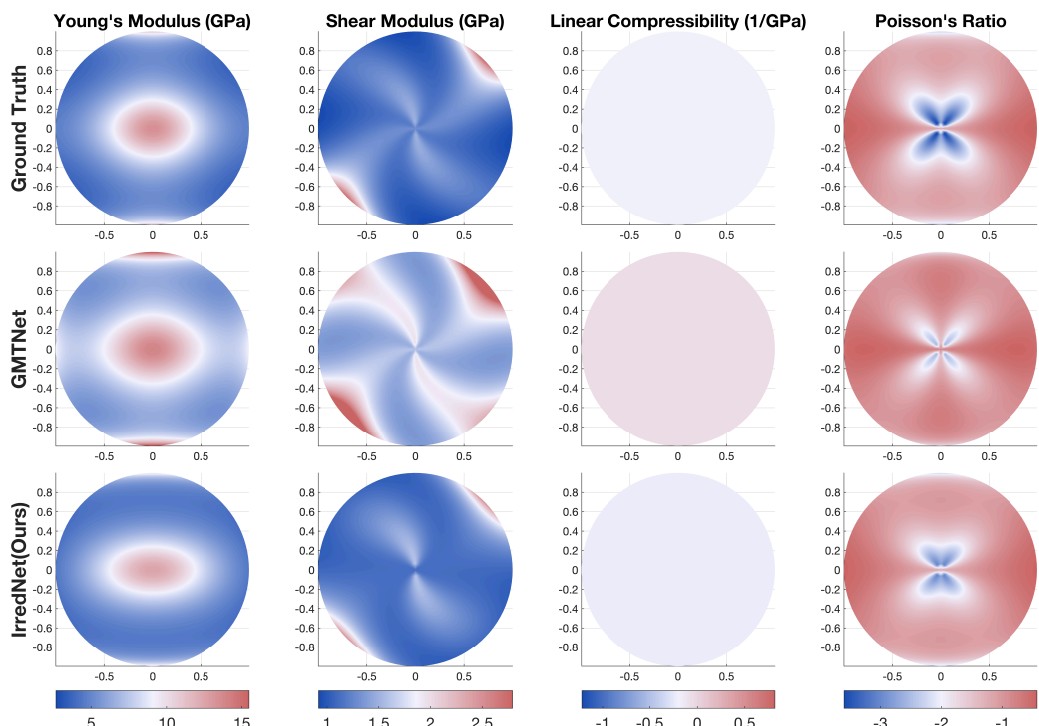

Figure 2: Directional elastic properties for a representative crystal. Columns show polar plots of Young's modulus (GPa), shear modulus (GPa), linear compressibility (1/GPa), and Poisson's ratio as functions of loading direction; rows correspond to ground truth (top), GMTNet (middle), and IrredNet (bottom). IrredNet more closely matches the ground-truth anisotropy and avoids the spurious artifacts visible in GMTNet.

## 4.4 ABLATION STUDIES

In this section, we evaluate the importance of symmetry and individual model components. To this end, we conduct ablation experiments on the dielectric dataset using several metrics, with results summarized in Table 4.4. For instance, on the Fnorm metric, incorporating irreducible decomposition delivers the first notable gain, reducing Fnorm from 4.17 (without decomposition) to 3.61. Notably, the piezoelectric $\ell_{\max}$ accuracy in predicting zero-elements and equality-elements improves significantly with this addition. However, when neither irreducible decomposition nor symmetry constraints are applied, the model struggles to capture symmetry accurately, leading to poor predictions for these elements. Only when both

Table 4: Ablation studies on the dielectric tensor dataset. Each row toggles components of IrredNet: Irred. Decomp. (irreducible-basis output parameterization), Sym. Cons. (exact symmetry constraints), Gate (gated equivariant convolutions), and Norm (tensor-norm prediction head). We report Fnorm (↓), , EwT (↑), and zero/equality accuracy (↑), showing that combining irreducible decomposition with symmetry constraints and gating yields the best overall performance and perfect symmetry compliance.

| Irred. Decomp. | Sym. Cons. | Gate | Norm | Fnorm (↓) | EwT 25% (↑) | EwT 10% (↑) | EwT 5% (↑) | Zero (↑) | Equality (↑) |
|---|---|---|---|---|---|---|---|---|---|
| ✗ | ✗ | ✓ | ✓ | 4.17 | 72.1% | 40.1% | 14.9% | 0% | 0% |
| ✓ | ✗ | ✓ | ✓ | 3.61 | 81.4% | 42.4% | 24.1% | 84.9% | 89.7% |
| ✓ | ✓ | ✗ | ✗ | 3.44 | 78.3% | 43.3% | 26.0% | 100% | 100% |
| ✓ | ✓ | ✓ | ✗ | 3.33 | 83.9% | 51.1% | 29.7% | 100% | 100% |
| ✓ | ✓ | ✓ | ✓ | **3.23** | **85.8%** | **53.2%** | **29.8%** | **100%** | **100%** |

irreducible decomposition and high-order tensor-specific symmetry constraints are applied can the model fully guarantee symmetry compliance, further reducing the Fnorm to 3.33. This demonstrates the effectiveness of our symmetry-constrained irreducible decomposition approach. Finally, experiments with the gate and normalization layers show that both contribute positively to the prediction accuracy of the model, with the best performance achieved when all components are used, resulting in a final Fnorm of 3.23.

### 4.5 Analysis of high-order tensor

To further validate our model's practical utility, we applied it to four material properties highly correlated with the elastic tensor, including Young's modulus, shear modulus, linear compressibility, and Poisson's ratio, which are widely used in materials engineering. Using HgIBr as a case study, we first computed its elastic tensor via the Elate tool Gaillac et al. (2016) and derived the four physical quantities. We then visualized and compared two sets of predictions against the ground truth in Figure 2: (1) ground truth, (2) GMTNet, and (3) IrredNet. The results show that our model's elastic tensor predictions and the associated physical quantities—achieve a higher degree of spatial-orientation consistency with the DFT reference than GMTNet, further confirming the superiority and robustness of our approach.

## 5 Conclusion

We address the problem of enforcing strict symmetry in high-order tensor predictions for crystals. We introduce IrredNet, which guarantees exact compliance by predicting symmetry-constrained irreducible components: we prove that any high-order tensor admits an SO(3) irreducible decomposition and provide an algorithm to enumerate its basis tensors, ensuring symmetry by construction. Experiments confirm exact symmetry preservation and high accuracy on complex tensors and related applications.

## 6 Reproducibility statement

We evaluate on publicly available datasets and standard benchmarks; data sources, licenses, splits, and preprocessing steps are cited in the main text and documented in Appendix D. We fix random seeds, report results over multiple runs, and list relevant hyperparameters, training schedules, and evaluation metrics in Appendix D, together with ablation protocols and baselines. Theoretical assumptions and complete proofs for the irreducible decomposition and exact symmetry preservation are provided in the appendix, and we include checks for symmetry compliance and reconstruction consistency. Environment details (hardware and library versions) and configuration tables are also provided to facilitate replication. We will publicly release the full codebase, configuration files, and pretrained checkpoints upon acceptance to enable end-to-end reproduction of all results.

## 7 Ethics Statement

We adhere to the ICLR Code of Ethics. This study uses publicly available, non-sensitive materials datasets only; no human subjects or personal data are involved, so IRB approval is not applicable. We document dataset composition, report out-of-domain evaluations to mitigate bias, avoid hazardous applications, and disclose key compute details; we identify no conflicts of interest.

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

# A    Theory for high-order tensor irreducible decomposition under $\mathrm{SO}(3)$ group

## A.1    Proof for high-order tensor irreducible decomposition under $\mathrm{SO}(3)$ group

**Proposition 1 Restated**    Let $V$ be a three-dimensional Euclidean vector space. Any rank-$k$ $(k > 1)$ tensor $\mathcal{T}_{i_1 \cdots i_k} \in V^{\otimes k}$ admits a unique decomposition into irreducible representations (irreps) of $\mathrm{SO}(3)$ group labeled by integer spins $\ell$ as follows:

$$V^{\bigotimes k} \simeq \bigoplus_{\ell \in L_k} \left( \mathcal{S}_k^{(\ell)} \bigoplus \mathcal{A}_k^{(\ell)} \bigoplus \mathcal{M}_k^{(\ell)} \right),$$

where $L_k$ is the set of all distinct integer spins $\ell$ that appear in the decomposition of $V^{\otimes k}$. $\mathcal{S}_k^{(\ell)}$, $\mathcal{A}_k^{(\ell)}$, and $\mathcal{M}_k^{(\ell)}$ denote the symmetric traceless, the antisymmetric, and the mixed-symmetry components(including symmetric with trace ) transforming under spin-$\ell$ irreps, respectively.

**Proof**    We aim to prove that any rank-$k$ $(k > 1)$ tensor $\mathcal{T}_{i_1 \cdots i_k} \in V^{\otimes k}$ (where V is a three-dimensional Euclidean vector space) admits a unique decomposition into irreducible representations (irreps) of the $\mathrm{SO}(3)$ group. These irreps are labeled by integer spins $\ell$ and can be categorized by their permutation symmetry properties.

Specifically, in **Step 1**, we establish the theoretical foundation for decomposing rank-k tensors under the $\mathrm{SO}(3)$ symmetry group by considering the irreducible representations (irreps) associated with $\mathrm{SO}(3)$ and their spin labels. **Step 2** focuses on the role of the symmetric group $S_k$ and Schur-Weyl duality, which enables the decomposition of the tensor space into subspaces defined by both $\mathrm{SO}(3)$ and $S_k$. **Step 3** addresses the condition of irreducibility under $\mathrm{SO}(3)$ by ensuring that all trace components are removed from the tensors, maintaining the required symmetry. In **Step 4**, we classify the irreducible components based on their permutation symmetry, dividing them into fully symmetric traceless, fully antisymmetric, and mixed symmetry subspaces. Finally, **Step 5** provides the explicit procedure for decomposing the tensor space, using Young symmetrizers (Nazarov, 1997; Brini & Teolis, 1989) and projection operations to achieve a unique decomposition of the space into traceless irreducible subspaces.

### Step 1: Representation-Theoretic Foundations

The group $\mathrm{SO}(3)$ is a compact Lie group Iserles et al. (2000); Chevalley (2018), and its finite-dimensional irreducible representations (irreps) are characterized by an integer spin (often denoted $\ell$), designated as $D(\ell)$ with dimension $2\ell + 1$. The three-dimensional Euclidean vector space V itself is the fundamental representation space for $\mathrm{SO}(3)$, corresponding to spin $\ell=1$. The space of rank-$k$ tensors, $V^{\otimes k}$, forms a representation of $\mathrm{SO}(3)$ that transforms under the action of $g \in \mathrm{SO}(3)$. For $k > 1$, this representation is generally reducible, and our goal is to decompose it into a direct sum of $\mathrm{SO}(3)$ irreps $D(\ell)$. By the general theory of compact groups, this decomposition is unique up to isomorphism and the ordering of terms.

### Step 2: Role of the Symmetric Group $S_k$ and Schur-Weyl Duality

The symmetric group $S_k$ acts on $V^{\otimes k}$ by permuting the tensor indices. The actions of the $\mathrm{SO}(3)$ group (acting on the values at each index position) and the $S_k$ group (acting on the index positions themselves) commute. This commutativity allows for the use of Schur-Weyl duality Brundan & Kleshchev (2008); Doty & Hu (2009), which decomposes $V^{\otimes k}$ into subspaces associated with the joint representations of $S_k$ and $\mathrm{SO}(3)$. Specifically,

$$V^{\otimes k} \simeq \bigoplus_{\lambda} \mathcal{R}_\lambda \otimes \mathcal{P}_\lambda,$$

where $\lambda$ are Young diagrams for partitions of $k$ (with at most $\dim(V)=3$ rows), $\mathcal{P}_\lambda$ is the irrep of $S_k$, and $\mathcal{R}_\lambda$ carries a representation of $\mathrm{SO}(3)$. Importantly, $\mathcal{R}_\lambda$ itself is generally reducible under $\mathrm{SO}(3)$ and requires further decomposition.

### Step 3: Irreducibility under$\mathrm{SO}(3)$ and Tracelessness

An irreducible tensor under SO(3) must be traceless. This is because the operation of contracting any pair of tensor indices using the Kronecker delta $\delta_{ij}$ (which is proportional to the metric tensor in Euclidean space) is an SO(3)-invariant operation Klebanov & Tarnopolsky (2017). If a tensor possessed a nonzero trace, then the trace (a tensor of rank $k-2$) and its traceless counterpart would span SO(3)-invariant subspaces, implying the original tensor was reducible. Therefore, to obtain SO(3) irreps from the spaces $\mathcal{R}_\lambda$ (which have a defined permutation symmetry), all trace components must be systematically removed.

**Step 4: Classification of Irreducible Components based on Permutation Symmetry**

The irreducible subspaces $D^{(\ell)}$ obtained from $V^{\otimes k}$ are classified by the permutation symmetry (Young diagram $\lambda$) of the tensors that carry them, together with the traceless condition:

- **Fully Symmetric** ($\mathcal{S}_k^{(\ell)}$): These components arise from tensors of Young diagram $\lambda = [k]$ (a single row, i.e. fully symmetric tensors). By convention $\mathcal{S}_k^{(\ell)}$ denotes the highest–spin component, namely $D^{(k)}$ (spin $\ell = k$), a fully symmetric traceless tensor of rank $k$.

- **Fully Antisymmetric** ($\mathcal{A}_k^{(\ell)}$): These components arise from tensors of Young diagram $\lambda = [1^k]$ (a single column, i.e. fully antisymmetric tensors). In $\dim(V) = 3$ they are nonzero only for $k \leq 3$, yielding for example $D^{(1)}$ (axial vector) when $k = 2$ and $D^{(0)}$ (pseudoscalar) when $k = 3$.

- **Mixed Symmetry** ($\mathcal{M}_k^{(\ell)}$): This category includes all other SO(3) irreps $D^{(\ell)}$ found in $V^{\otimes k}$. They come from Young diagrams that are neither a single row without trace nor a single column (e.g. $\lambda = [k-1, 1]$ for $k \geq 3$), after imposing tracelessness. Consistent with the proposition's convention on traces of symmetric tensors, $\mathcal{M}_k^{(\ell)}$ also contains the lower–spin irreps $D^{(k-2j)}$ ($j > 0$) originating from the trace of the $[k]$ diagram once the leading $\mathcal{S}_k^{(k)}$ component is removed.

**Step 5: Explicit Decomposition, Projections, and Uniqueness**

The decomposition of the $V^{\otimes k}$ space into subspaces of specific permutation symmetry (corresponding to Young diagrams $\lambda$) can be achieved using Young symmetrizers $\mathbb{P}_\lambda$, which are linear combinations of permutation operators from $S_k$; examples include the full symmetrizer $\mathbb{P}_S$ and antisymmetrizer $\mathbb{P}_A$. Within each such symmetry-defined subspace $\mathcal{R}_\lambda$, further projection operations are necessary to remove all trace components, ensuring the resulting tensors are traceless with respect to all pairs of indices.

The resulting traceless tensors form the basis for the SO(3) irreps $D^{(\ell)}$. The specific spin value $\ell$ for each irreducible subspace can be confirmed by the eigenvalue equation for the total angular momentum operator squared, $J^2 \mathcal{T}^{(\ell)} = \ell(\ell+1)\mathcal{T}^{(\ell)}$. The decomposition of $V^{\otimes k}$ into a direct sum of SO(3) irreps $D^{(\ell)}$ is unique. The categorization into $\mathcal{S}_k^{(\ell)}$, $\mathcal{A}_k^{(\ell)}$, and $\mathcal{M}_k^{(\ell)}$ (adhering to the specific definitions where $\mathcal{S}_k^{(\ell)}$ is primarily the $\ell = k$ part from symmetric tensors) provides a complete partitioning of these irreps based on their permutation symmetry origins and the traceless condition.

**Proposition 2 Restated** Let $\mathcal{T}$ be a rank-$k$ tensor defined over a three-dimensional Euclidean space. Under decomposition into irreducible representations (irreps) of SO(3) group: When $\mathcal{T}$ satisfies additional index symmetries (e.g., major/minor symmetry, tracelessness, or antisymmetry), the set of allowed irreps forms a proper subset of the unconstrained case: $\mathrm{Irrep}(\mathcal{T}_{\mathrm{sym}}) \subsetneq \mathrm{Irrep}(\mathcal{T}_{\mathrm{unconstrained}})$.

**Proof** We prove the Proposition 2 based on the Schur's Lemma Chang & Skjelbred (1974); Serre et al. (1977).

**Schur's lemma**  Let $(\rho, V)$ and $(\rho', W)$ be two irreducible representations of a group $G$. If a linear map $\phi : V \to W$ satisfies:

$$\phi \circ \rho(g) = \rho'(g) \circ \phi \quad (\forall g \in G),$$

Then if $V$ and $W$ are inequivalent, then $\phi = 0$. If $V \simeq W$, then $\phi$ is a scalar multiple of the identity ($\phi = \lambda I$).

Let $\mathcal{T}$ be a rank-$k$ tensor in 3D Euclidean space. We rigorously prove that imposing symmetry constraints reduces the set of admissible irreducible representations (irreps) of SO(3) in its decomposition by Schur's lemma.

For a tensor space without symmetry constraints $\mathcal{T} = V^{\otimes k}$, its SO(3)-irreducible decomposition is:

$$\mathcal{T} \simeq \bigoplus_\ell m_\ell \mathcal{H}^{(\ell)}$$

where $\mathcal{H}^{(\ell)}$ is the spin-$\ell$ irrep. A symmetry constraint $\mathcal{S}$ induces a projection operator $\Pi_\mathcal{S} : \mathcal{T} \to \mathcal{T}_\mathcal{S}$ satisfying:

$$\Pi_\mathcal{S} \circ \rho(g) = \rho(g) \circ \Pi_\mathcal{S} \quad (\forall g \in \mathrm{SO}(3))$$

i.e., $\Pi_\mathcal{S}$ is an SO(3)-equivariant map. The projection $\Pi_\mathcal{S}$ decomposes into irreducible subspaces:

$$\Pi_\mathcal{S} = \bigoplus_\ell \Pi_\mathcal{S}^{(\ell)}, \quad \Pi_\mathcal{S}^{(\ell)} : \mathcal{H}^{(\ell)} \to \mathcal{H}^{(\ell)},$$

where by Schur's Lemma, each $\Pi_\mathcal{S}^{(\ell)}$ must be a scalar multiple of the identity: $\Pi_\mathcal{S}^{(\ell)} = \lambda_\ell I$. If a constraint $\mathcal{S}$ is incompatible with $\mathcal{H}^{(\ell)}$, then $\Pi_\mathcal{S}^{(\ell)} = 0$ ($\lambda_\ell = 0$), meaning this irrep cannot appear in $\mathcal{T}_\mathcal{S}$. Conversely, compatibility implies $\lambda_\ell \neq 0$. We therefore conclude that the symmetry constraint $\mathcal{S}$ excludes certain irreducible representations from the decomposition, resulting in a reduced set of allowable irreps: $\mathcal{I} = \{\ell \mid \lambda_\ell \neq 0\} \subsetneq \{0, 1, \ldots, k\}$.

# B  EXAMPLES OF HIGH-ORDER TENSOR IRREDUCIBLE DECOMPOSITION UNDER SO(3) GROUP

In this section, we present three illustrative examples demonstrating the irreducible decomposition of tensors of order 2 to 4 under the SO(3) group, based on **Proposition 1** and **Proposition 2**. We have validated the correctness of these examples through a series of references Itin & Reches (2022); Itin (2018); Browaeys & Chevrot (2004); Kolda & Bader (2009); Brachat et al. (2010).

## B.1  IRREDUCIBLE DECOMPOSITION OF 2ND-ORDER TENSOR DECOMPOSITION

**1. Unconstrained Case**  Any rank-2 tensor $T_{ij}$ in 3D space can be decomposed into symmetric and antisymmetric parts, which further reduce into irreducible representations (irreps) of the rotation group SO(3).

$$T_{ij} \simeq \underbrace{1 \times \mathbf{1}}_{\ell=0} \oplus \underbrace{1 \times \mathbf{3}}_{\ell=1} \oplus \underbrace{1 \times \mathbf{5}}_{\ell=2}.$$

(1) Symmetric Traceless Component ($\mathcal{S}_2^{(\ell)}$)

$\mathcal{S}_2^{(2)}$: Traceless part (**5** independent modes)

$$\mathcal{S}_2^{(2)} : \quad T_{(ij)} - \frac{1}{3}\delta_{ij}T_{kk}, \quad \text{where } T_{(ij)} = \frac{1}{2}(T_{ij} + T_{ji}).$$

(2) Antisymmetric Component ($\mathcal{A}_2^{(\ell)}$):

$(\mathcal{A}_2^{(1)})$: The skew-symmetric part maps to a pseudovector via the Levi-Civita symbol (**3** independent modes):

$$\mathcal{A}_2^{(1)}: \quad T_{[ij]} = \frac{1}{2}(T_{ij} - T_{ji}), \quad \text{equivalent to } \mathbf{v}_k = \epsilon_{ijk} T_{[ij]}.$$

(3) Symmetric Trace Components $(\mathcal{S}_2^{(\ell)})$ (can also be formulated as $\mathcal{M}_2^{(\ell)}$ based on **Proposition 1**)

The scalar (isotropic) part formed by the trace (**1** independent mode):

$$\mathcal{S}_2^{(0)}: \quad \frac{1}{3}\delta_{ij} T_{kk}, \quad \text{where } T_{kk} = \delta_{ij} T_{ij}.$$

**Dimension Verification:** $1 \times 1 + 1 \times 3 + 1 \times 5 = 9 = 3^2$ (matches rank-2 tensor in 3D).

**2. Symmetry constraints** In the dielectric tensor, due to the symmetry $T_{ij} = T_{ji}$, its antisymmetric part automatically disappears.

$$T_{ij} \simeq \underbrace{1 \times \mathbf{1}}_{\ell=0} \oplus \underbrace{1 \times \mathbf{5}}_{\ell=2}.$$

B.2 IRREDUCIBLE DECOMPOSITION OF 3RD-ORDER TENSOR DECOMPOSITION

**1. Unconstrained Case** $T_{ijk}$ decomposes into irreducible representations (irreps) of SO(3) labeled by angular momentum $L$:

$$T_{ijk} \simeq \underbrace{1 \times \mathbf{1}}_{\ell=0} \oplus \underbrace{3 \times \mathbf{3}}_{\ell=1} \oplus \underbrace{2 \times \mathbf{5}}_{\ell=2} \oplus \underbrace{1 \times \mathbf{7}}_{\ell=3},$$

(1) Symmetric Traceless Components $(\mathcal{S}_3^{(\ell=3)})$

$\ell = 3$ (Octupole): Fully symmetric traceless part (with **7** independent modes):

$$\mathcal{S}_{ijk}^{(3)}: \quad T_{(ijk)} - \frac{1}{5}\left(\delta_{ij}\langle T\rangle_k + \delta_{ik}\langle T\rangle_j + \delta_{jk}\langle T\rangle_i\right),$$

$$\langle T\rangle_k \equiv T_{ppk}.$$

(2) Antisymmetric Components $(\mathcal{A}_3^{(\ell)})$

$\mathcal{A}_3^{(\ell=0)}$ with only **1** independent mode using Levi-Civitta contraction:

$$\mathcal{A}_{ijk}^{(\ell=0)}: \quad \frac{1}{6}\epsilon_{ijk}\left(\epsilon_{pqr} T_{pqr}\right).$$

(3) Mixed-symmetry Components $(\mathcal{M}_3^{(\ell=1)}$ and $\mathcal{M}_3^{(\ell=2)})$

Type-I (Symmetric in 2 indices):

$$T_{(ij)k} - \frac{1}{3}\delta_{ij}\langle T\rangle_k$$

Decompose $T_{(ij)k}$ into:

- Trace: $\langle T\rangle_k = T_{ppk} \to \ell = 1$ (**3** independent modes).
- Traceless: Project out $\ell = 2$ (**5** independent modes).

Type-II (Antisymmetric in 2 indices):

$$T_{[ij]k} - \frac{1}{3}\epsilon_{ijk}(\epsilon_{pqr} T_{[pq]r})$$

$T_{[ij]k}$ splits into:

- Vector: $v_k = \epsilon_{ijm} T_{[ij]k} \to \ell = 1$ (**3** independent modes).
- Traceless: Constraints reduce to $\ell = 2$ (**5** independent modes).

Total Mixed-Symmetric:

$$\mathcal{M}_3 = \underbrace{3 \times \mathbf{3}}_{\ell=1} \oplus \underbrace{2 \times \mathbf{5}}_{\ell=2}.$$

**Dimension Verification:** $1 \times 1 + 3 \times 3 + 2 \times 5 + 1 \times 7 = 27 = 3^4$. (matches rank-3 tensor in 3D)

**2. Symmetry Constraints** When $T_{ijk}$ is symmetric in its last two indices ($T_{ijk} = T_{ikj}$ in piezoelectric tensor $\epsilon_{ijk}$), the decomposition simplifies dramatically:

$$T_{ijk} \simeq \underbrace{2 \times \mathbf{3}}_{\ell=1} \oplus \underbrace{1 \times \mathbf{5}}_{\ell=2} \oplus \underbrace{1 \times \mathbf{7}}_{\ell=3},$$

reducing the independent components from 27 to 18.

### B.3 Irreducible decomposition of 4th-order tensor decomposition

**1. Unconstrained Case** For a rank-4 tensor $C_{ijkl} \in V^{\otimes 4}$, the decomposition is explicitly constructed as follows:

$$T_{ijkl} \simeq \underbrace{3 \times \mathbf{1}}_{\ell=0} \oplus \underbrace{6 \times \mathbf{3}}_{\ell=1} \oplus \underbrace{6 \times \mathbf{5}}_{\ell=2} \oplus \underbrace{3 \times \mathbf{7}}_{\ell=3} \oplus \underbrace{1 \times \mathbf{9}}_{\ell=4},$$

(1) Symmetric Components ($\mathcal{S}_4^{(\ell)}$)

$\ell = 4$ (Hexadecapole): Fully symmetric and traceless part.(only 1 independent mode), :

$$\mathcal{S}_4^{(4)} : \quad C_{(ijkl)} - \text{lower-order traces}.$$

This part is constructed by using the symmetrization operator $\mathcal{P}_4$ and subtracting all possible traces (e.g., $\delta_{ij}$ terms)

(2) Antisymmetric Components ($\mathcal{A}_4^{(\ell)}$)

In 3D, all $\mathcal{A}_4^{(\ell)}$ vanish because there is no nonzero fully antisymmetric rank-4 tensor (the Levi-Civita symbol $\epsilon_{ijk}$ only supports 3 indices).

(3) Mixed-Symmetry Components ($\mathcal{M}_4^{(\ell)}$)

$\ell = 0$ (Scalar): Independent modes are fully contracted scalars (can also be formulated as $\mathcal{S}_4^{(0)}$ :

$$\mathcal{M}_4^{(0)} : \quad C_{iijj}, \ C_{ijij}, \ C_{ijji} \quad \text{(3 independent scalars)}.$$

$\ell = 1$ (Vector): Constructed via contractions with $\epsilon_{mjk}$, yielding 6 independent vector modes:

$$\mathcal{M}_4^{(1)} : \quad V_i^{(n)} = \epsilon_{mjk} C_{ijkl}, \quad n = 1, \ldots, 6.$$

$\ell = 2$ (Quadrupole): Traceless symmetric tensors (6 independent modes):

$$\mathcal{M}_4^{(2)} : \quad C_{ijkl} - \text{fully symmetric part} - \text{scalar traces}.$$

$\ell = 3$ (Octupole): Higher-order traceless projections (3 independent modes):

$$\mathcal{M}_4^{(3)} : \quad \text{Project out contributions from } \ell = 0, 2, 4 \text{ using } \mathcal{P}_3.$$

**Dimension Verification:** $3 \times 1 + 6 \times 3 + 6 \times 5 + 3 \times 7 + 1 \times 9 = 81 = 3^4$. matches rank-4 tensor in 3D

**2. Symmetry Constraints** When applied major symmetry ($C_{ijkl} = C_{klij}$) and minor symmetry ($C_{ijkl} = C_{jikl} = C_{ijlk}$) on 4th-order tensor, the decomposition components are reduced:

$$T_{ijkl} \simeq \underbrace{2 \times \mathbf{1}}_{\ell=0} \oplus \underbrace{2 \times \mathbf{5}}_{\ell=2} \oplus \underbrace{1 \times \mathbf{9}}_{\ell=4},$$

## C  DETAILS OF THE PHYSICAL PRINCIPLES IN THE PROPERTIES OF HIGH-ORDER TENSORS

### C.1  DIELECTRIC

Dielectric tensor is calculated by $\mathbf{D} = \epsilon \mathbf{E}$, which can be written as:

$$\begin{pmatrix} D_x \\ D_y \\ D_z \end{pmatrix} = \begin{pmatrix} \varepsilon_{xx} & \varepsilon_{xy} & \varepsilon_{xz} \\ \varepsilon_{yx} & \varepsilon_{yy} & \varepsilon_{yz} \\ \varepsilon_{zx} & \varepsilon_{zy} & \varepsilon_{zz} \end{pmatrix} \begin{pmatrix} E_x \\ E_y \\ E_z \end{pmatrix}$$

where $\mathbf{E} \in \mathbb{R}^3$ is an externally applied electric field and $\mathbf{D}$ is the resultant electric displacement field $\mathbf{D} \in \mathbb{R}^3$

Due to the symmetry $\varepsilon_{ij} = \varepsilon_{ji}$, i.e., $\varepsilon_{xy} = \varepsilon_{yx}$, $\varepsilon_{xz} = \varepsilon_{zx}$, etc., the antisymmetric components are reduced to zero, leaving six independent modes, as discussed in Appendix B.1.

### C.2  PIEZOELECTRIC

Piezoelectric Tensor is calculated by

$$e_{ijk} = \left( \frac{\partial D_i}{\partial \varepsilon_{jk}} \right)_E = - \left( \frac{\partial \sigma_{jk}}{\partial E_i} \right)_\epsilon$$

where $\mathbf{D} \in \mathbb{R}^3, \mathbf{E} \in \mathbb{R}^3$ is the electric displacement tensor and the electric field, $\boldsymbol{\sigma} \in \mathbb{R}^{3\times3}, \boldsymbol{\epsilon} \in \mathbb{R}^{3\times3}$ is the second-order elastic stress tensor and strain tensor.

Due to the symmetry of the stress tensor $\boldsymbol{\sigma}_{ij} = \boldsymbol{\sigma}_{ij}$, the piezoelectric tensor satisfies the symmetry relation:

$$\mathbf{e}^{ijk} = \mathbf{e}^{ikj}$$

In general, such a pair-symmetric tensor has 18 independent components which are corresponding to the dimensions of Appendix B.2. Formally, the piezoelectric tensor can also be forumulated by Voigt notation as: The symmetry

$$\mathbf{e} = \begin{pmatrix} e_{11} & e_{12} & e_{13} & e_{14} & e_{15} & e_{16} \\ e_{21} & e_{22} & e_{23} & e_{24} & e_{25} & e_{26} \\ e_{31} & e_{32} & e_{33} & e_{34} & e_{35} & e_{36} \end{pmatrix}$$

### C.3  ELASTIC

The elastic tensor (or elastic stiffness tensor) $C_{ijkl}$ is a fourth-rank tensor that quantifies the linear relationship between strain $\epsilon_{kl}$ and stress $\sigma_{ij}$ in a solid material under small deformations (Hooke's law). It fully characterizes the stiffness (resistance to deformation) of an elastic material. According to the principle of elastic tensor, here are two definitions:

(1) **Stress-Strain Relationship** In the linear elasticity regime, the constitutive equation (generalized Hooke's law) is:

$$\sigma_{ij} = C_{ijkl}\epsilon_{kl}$$

where $\sigma_{ij}$ = Cauchy stress tensor (force per unit area, symmetric: $\sigma_{ij} = \sigma_{ji}$) $\epsilon_{kl}$ = infinitesimal strain tensor (symmetric: $\epsilon_{kl} = \epsilon_{lk}$) $C_{ijkl}$ = elastic stiffness tensor, with 81 components (reducible to 21 independent components due to symmetry).

(2) **Strain Energy Density** The elastic tensor is derived from the strain energy density $\Psi$ (energy per unit volume stored due to deformation), which is quadratic in strain for a linear elastic material:

$$\Psi = \frac{1}{2} C_{ijkl} \epsilon_{ij} \epsilon_{kl}$$

Stress is obtained from the thermodynamic derivative:

$$\sigma_{ij} = \frac{\partial \Psi}{\partial \epsilon_{ij}} = C_{ijkl} \epsilon_{kl}$$

The elastic tensor can also be defined as the second derivative of energy with respect to strain:

$$C_{ijkl} = \frac{\partial^2 \Psi}{\partial \epsilon_{ij} \partial \epsilon_{kl}}$$

Due to symmetry ( $\sigma_{ij} = \sigma_{ji}$, $\epsilon_{kl} = \epsilon_{lk}$ ), the tensor is compressed into a $6 \times 6$ stiffness matrix $\mathbf{C}_{\alpha\beta}$:

$$
\begin{bmatrix} \sigma_1 \\ \sigma_2 \\ \sigma_3 \\ \sigma_4 \\ \sigma_5 \\ \sigma_6 \end{bmatrix}
=
\begin{bmatrix}
C_{11} & C_{12} & C_{13} & C_{14} & C_{15} & C_{16} \\
C_{12} & C_{22} & C_{23} & C_{24} & C_{25} & C_{26} \\
C_{13} & C_{23} & C_{33} & C_{34} & C_{35} & C_{36} \\
C_{14} & C_{24} & C_{34} & C_{44} & C_{45} & C_{46} \\
C_{15} & C_{25} & C_{35} & C_{45} & C_{55} & C_{56} \\
C_{16} & C_{26} & C_{36} & C_{46} & C_{56} & C_{66}
\end{bmatrix}
\begin{bmatrix} \epsilon_1 \\ \epsilon_2 \\ \epsilon_3 \\ \epsilon_4 \\ \epsilon_5 \\ \epsilon_6 \end{bmatrix}
$$

where stress and strain are in Voigt Notation: (1) $\sigma_1 = \sigma_{11}$, $\sigma_4 = \sigma_{23}$ (shear components) (2) $\epsilon_1 = \epsilon_{11}$, $\epsilon_4 = 2\epsilon_{23}$ (engineering shear strain). Considering the whole symmetry of elastic tensor ($C_{ijkl} = C_{klij} = C_{jikl} = C_{ijlk}$), 36 components can be further reduced into 21 components, which correspond to B.3:

$$
C^{ijkl} =
\begin{bmatrix}
C^{1111} & C^{1122} & C^{1133} & C^{1123} & C^{1131} & C^{1112} \\
* & C^{2222} & C^{2233} & C^{2223} & C^{2231} & C^{2212} \\
* & * & C^{3333} & C^{3323} & C^{3331} & C^{3312} \\
* & * & * & C^{2323} & C^{2331} & C^{2312} \\
* & * & * & * & C^{3131} & C^{3112} \\
* & * & * & * & * & C^{1212}
\end{bmatrix}
$$

$$
=
\begin{bmatrix}
C^{11} & C^{12} & C^{13} & C^{14} & C^{15} & C^{16} \\
* & C^{22} & C^{23} & C^{24} & C^{25} & C^{26} \\
* & * & C^{33} & C^{34} & C^{35} & C^{36} \\
* & * & * & C^{44} & C^{45} & C^{46} \\
* & * & * & * & C^{55} & C^{56} \\
* & * & * & * & * & C^{66}
\end{bmatrix}
$$

# D  PARAMETER SETTING

## D.1  DETAILS OF BASELINES

**MEGNET**   Implementations of MEGNet Chen et al. (2019). Following the original paper, we use three layers of MEGNet message passing with the same feature dimensions as mentioned in the paper. We train MEGNet for 200 epochs using Huber loss with a learning rate of 0.001 and AdamW optimizer with $10^{-5}$ weight decay. The same polynomial learning rate decay scheduler as our model implementation is used. To predict a tensor matrix of shape $3 \times 3$, we change the original output dimension from one to nine.

**ETGNN** Following ETGNN Zhong et al. (2022), we use four DimeNet++ Gasteiger et al. (2020) layers with hidden dimension 128 and ELU activation function to serve as the invariant message-passing network. Since the code of ETGNN is not publically accessible, we implement DimeNet++ layers following the code provided by GMTNet. We train ETGNN for 200 epochs using Huber loss with a learning rate of 0.001 and AdamW optimizer with $10^{-5}$ weight decay. The same polynomial learning rate decay scheduler as our model implementation is used.

**GMTNet** Following original paper Yan et al. (2024b), we use 2 layers of node invariant feature updating and 3 layers of equivariant message passing for all tasks including dielectric, piezoelectric, and elastic tensors. Learning rate of 0.001, epoch number of 200, and batch size of 64 are used for dielectric, piezoelectric, and elastic tensors. The radius of construction crystal graphs is determined by the 16-th nearest neighbor.

### D.2 Details of IrredNet

Our model is implemented based on the MACE Batatia et al. (2022) and GMTNet repositories. In the graph construction block, we set the radius to 5 and use 20 basis functions of the Bessel function for encoding edge length. The angular momentum number $\ell = 3$ (for dielectric), $\ell = 3$ (for piezoelectric), and $\ell = 4$ (for elastic) are used to encode the edge directions. Additionally, we employ four spherical harmonic convolution layers, along with gates and batch normalization, to learn the node features.Learning rate of 0.001, epoch number of 200, and batch size of 64 are used for dielectric, piezoelectric, and elastic tensors. All experiments were conducted on a computing platform with four NVIDIA RTX 3090 GPUs, running Ubuntu 22.04, an Intel Xeon Gold 5120 CPU at 2.20 GHz, and 512 GB of system memory. Our code will be released when paper is publicly available.

## E Analysis of high-order tensor properties

### E.1 Analysis of Elastic Tensor

The elastic stiffness tensor $C_{ijkl}$ and the compliance tensor $S_{ijkl}$ are two fundamental fourth-rank tensors in linear elasticity theory. They are inverse tensors of each other and together describe the stress-strain response of a material. In Voigt notation, **C** and **S** become $6 \times 6$ matrices, with $\mathbf{S} = \mathbf{C}^{-1}$. Formally, it's more common to use compliance tensor $S_{ijkl}$ for calculating relevant properties Browaeys & Chevrot (2004); Cowin (1985).

**Youngs' modulus** $E(n)$ Young's modulus describes the material's stiffness in the direction defined by the unit vector n. It is the inverse of the compliance tensor contracted along the direction of the applied stress.

$$E(\mathbf{n}) = \left( S_{ijkl}\, n_i n_j n_k n_l \right)^{-1}.$$

**Poisson's ratio** $\nu(n, m)$ Poisson's ratio characterizes the material's deformation in the lateral directions when stretched or compressed along the direction defined by the unit vector n. It is the ratio of lateral strain to axial strain.

$$\nu(\mathbf{n}, \mathbf{m}) = -\frac{S_{ijkl} n_i n_j m_k m_l}{S_{pqrs} n_p n_q n_r n_s}$$

**Linear compressibility** $\beta(n)$ Linear compressibility measures the material's response to uniform compression, defined as the negative of the inverse of the bulk modulus $P$, using the compliance tensor to calculate the volumetric strain.

$$\beta(\mathbf{n}) = -\frac{1}{P}\, \varepsilon^{(\mathbf{n})} = M_{ij}\, n_i\, n_j = S_{ijkl}\, \delta_{kl}\, n_i\, n_j = S_{ijkk}\, n_i\, n_j.$$

**Shear modulus** $G(n, m)$ The shear modulus $G(n, m)$ describes the material's resistance to shear deformation. It is calculated from the compliance tensor in the direction of the applied

shear stress and strain.

$$G(\mathbf{n}, \mathbf{m}) = \frac{\tau}{\gamma} = \frac{1}{4\, S_{ijkl}\, n_i\, m_j\, n_k\, m_l} \,.$$

# F    Supplementary pseudo code analysis

## F.1    Process of GenerateSO3Irreps

The process begins with the construction of an unconstrained decomposition map, $\mathcal{M}_0$. The theoretical basis for this step is rooted in the representation theory of the general linear group GL(3) and the symmetric group SO(3). This relies on key group-theoretical tools: first, Schur-Weyl Duality Brachat et al. (2010) is employed to establish a correspondence between the tensor's permutation symmetries and the GL(3) irreducible representations indexed by Young Diagrams. Subsequently, Branching Rules are applied to decompose each GL(3) representation into a direct sum of SO(3) representations upon restriction to the SO(3) subgroup Gusev et al. (2015); Costa & Hansen (2015). This systematic approach, by iterating through all valid Young diagrams, automatically accounts for the contributions from all of the tensor's symmetry components, including the totally symmetric (corresponding to single-row diagrams), totally antisymmetric (single-column diagrams), and mixed symmetries (all other diagram shapes).

---

**Algorithm 2** Supplementary Pseudo Code of Function HookLengthFormula

---

    **Input:** An integer partition $\lambda = [\lambda_1, \lambda_2, \dots]$
    **Output:** The dimension of the $S_k$ irreducible representation for $\lambda$
1: **function** HookLengthFormula($\lambda$)
2:     $k \leftarrow \sum_{i=1}^{\text{length}(\lambda)} \lambda_i$
3:     **if** $k = 0$ **then return** 1
4:     **end if**
5:     $k_{\text{factorial}} \leftarrow k!$
6:     hook_product $\leftarrow 1$
7:     **for** $i \leftarrow 1$ to length($\lambda$) **do**          ▷ Iterate through the rows of the Young diagram
8:         **for** $j \leftarrow 1$ to $\lambda_i$ **do**       ▷ Iterate through the columns of the Young diagram
9:             boxes_right $\leftarrow \lambda_i - j$
10:            boxes_below $\leftarrow 0$
11:            **for** $m \leftarrow i + 1$ to length($\lambda$) **do**
12:                **if** $\lambda_m \geq j$ **then**
13:                   boxes_below $\leftarrow$ boxes_below $+ 1$
14:                **end if**
15:            **end for**
16:            hook_length $\leftarrow$ boxes_right $+$ boxes_below $+ 1$
17:            hook_product $\leftarrow$ hook_product $\times$ hook_length
18:         **end for**
19:     **end for**
20:     dimension $\leftarrow k_{\text{factorial}}/$hook_product
21:     **return** round(dimension)
22: **end function**

---

## F.2    Complexity analysis of irreducible decomposition

We also conduct analysis on complexity of irreducible decomposition of high-order tensor. The algorithm consists of two main parts: 1. $\mathcal{M}_0 \leftarrow$ GenerateIrreps$k$ : Generates the unconstrained decomposition, which is the most computationally expensive part. 2. $\mathcal{M}_{final} \leftarrow$ ApplySymmetry$\mathcal{M}_0, G_{sym}$.

---

**Algorithm 4** Supplementary Pseudo Code of Function GENERATEPARTITIONS

---

**Input:** Integers $p = \lambda_1 - \lambda_2$, $q = \lambda_2 - \lambda_3$
**Output:** A map $M_{\text{branch}}$ from an SO(3) irrep $L$ to its multiplicity
1: **function** BRANCHRULESU3TOSO3($p, q$)
2:     $M_{\text{branch}} \leftarrow$ new Map()
3:     $\lambda \leftarrow p$
4:     $\mu \leftarrow q$
5:     **for** $K \leftarrow \mu$ down to 0 step $-2$ **do**
6:         **if** $K = 0$ **then**
7:             **for** $L \leftarrow \lambda$ down to 0 step $-2$ **do**
8:                 **if** $M_{\text{branch}}$ has key $L$ **then**
9:                     $M_{\text{branch}}[L] \leftarrow M_{\text{branch}}[L] + 1$
10:                 **else**
11:                     $M_{\text{branch}}[L] \leftarrow 1$
12:                 **end if**
13:             **end for**
14:         **else**
15:             **for** $L \leftarrow K$ to $K + \lambda$ **do**
16:                 **if** $M_{\text{branch}}$ has key $L$ **then**
17:                     $M_{\text{branch}}[L] \leftarrow M_{\text{branch}}[L] + 1$
18:                 **else**
19:                     $M_{\text{branch}}[L] \leftarrow 1$
20:                 **end if**
21:             **end for**
22:         **end if**
23:     **end for**
24:     **return** $M_{\text{branch}}$
25: **end function**

---

**Input:** Integer $k$, and max number of rows/parts max_rows
**Output:** A list of all partitions of $k$ with at most max_rows parts
1: **function** GENERATEPARTITIONS($k$, max_rows)
2:     all_parts $\leftarrow$ new List()
3:     cur_part $\leftarrow$ new List()
4:     **if** $k > 0$ **then**
5:         REFINDPARTS($k$, max_rows, $k$, cur_part, all_parts)
6:     **end if**
7:     **return** all_parts
8: **end function**
9: **procedure** REFINDPARTS($k$, max_parts, start_num, cur_part, all_parts)
10:     **if** $k = 0$ **then**                ▷ Valid
11:         Add a copy of cur_part to all_parts
12:         **return**
13:     **end if**
14:     **if** $k < 0$ **or** max_parts $= 0$ **then**     ▷ Invalid partition
15:         **return**
16:     **end if**
17:     **for** $i \leftarrow \min(k, \text{start\_num})$ down to 1 **do**     ▷ Iterate
18:         Add $i$ to cur_part
19:         REFINDPARTS($k - i$, max_parts-1, $i$, cur_part, all_parts)
20:         Remove the last element ($i$) from cur_part
21:     **end for**
22: **end procedure**

---

**Step 1: GeneratePartitions(k, max_rows=3)** The number of integer partitions of $k$ into at most 3 parts, $P_3(k)$, is on the order of $O(k^2)$. The time and space required to generate these partitions are both $O(k^2)$.

**Main Loop** This loop iterates $P_3(k) \approx O(k^2)$ times. The complexity of the loop body is as follows:

- **Step 2** (HOOKLENGTHFORMULA): Time complexity is $O(k)$.
- **Step 3** (BRANCHSU3TOSO3): The complexity is approximately $O(p \times q)$, which in the worst case is $O(k^2)$.
- **Step 4** (Accumulation): This loop runs $O(k^2)$ times, resulting in a complexity of $O(k^2)$.

**Summary for GENERATESO3IRREPS** The complexity as summarized as follows:

- **Time Complexity:** The main loop runs $O(k^2)$ times, and the most expensive step within it is $O(k^2)$. The total time complexity is $O(k^2) \times O(k^2) = O(k^4)$.
- **Space Complexity:** Dominated by storing the $O(k^2)$ partitions, the space complexity is $O(k^2)$.

2. COMPLEXITY ANALYSIS OF APPLYSYMMETRY($\mathcal{M}_0$, $G_{\text{SYM}}$)

This function filters the unconstrained decomposition map $\mathcal{M}_0$.

**Loop Iterations** The size of $\mathcal{M}_0$ is at most $k + 1$ (since $\ell_{max} = k$), so the main loop runs $O(k)$ times.

**Function Call** The ISCOMPATIBLE function's complexity depends on the number of conjugacy classes, $|C_{G_{\text{sym}}}|$, which is a constant with respect to $k$. Thus, its complexity is $O(|C_{G_{\text{sym}}}|)$.

**Summary for APPLYSYMMETRY** The complexity as summarized as follows:

- **Time Complexity:** $O(k) \times O(|C_{G_{\text{sym}}}|) = O(k \cdot |C_{G_{\text{sym}}}|)$.
- **Space Complexity:** A new map of size at most $O(k)$ is created, so the space complexity is $O(k)$.

FINAL CONCLUSION

By combining the analysis of all parts, we can determine the overall complexity of the algorithm:

- **Total Time Complexity:** $T(k) = O(k^4) + O(k \cdot |C_{G_{\text{sym}}}|)$. Since $k^4$ is the dominant term, the final time complexity is $\boldsymbol{O(k^4)}$.
- **Total Space Complexity:** $S(k) = O(k^2) + O(k)$. Since $k^2$ is the dominant term, the final space complexity is $\boldsymbol{O(k^2)}$.

The computational cost of the algorithm is primarily driven by the GENER-ATESO3IRREPS function. In practice, during training this computation has minimal impact on overall cost; for each tensor it only needs to be performed once upfront.

## G COMPLEXITY ANALYSIS IN ALGORITHM 1

**Complexity of basis generation.** Algorithm 1 consists of two stages: (i) the unconstrained SO(3) irreducible decomposition via GENERATESO3IRREPS($k$) and (ii) the symmetry-compatibility filtering APPLYSYM($M_0, G_{\text{sym}}$), with an optional projector step to obtain explicit invariant bases. The function GENERATESO3IRREPS($k$) enumerates all partitions of the tensor rank $k$ with at most 3 parts and applies the SU(3)→SO(3) branching rules (Appendix F). The number of such partitions grows as

$$p_3(k) = \Theta(k^2),$$

and each branching call has quadratic cost in $k$, leading to an overall runtime

$$\mathcal{O}(k^4)$$

and $\mathcal{O}(|L_k|) = \mathcal{O}(k)$ memory for storing the map $M_0[\ell]$, where $L_k$ is the set of distinct spins. The symmetry step ApplySym$(M_0, G_{\text{sym}})$ loops over all $\ell \in L_k$ and, for each spin, evaluates the O(3) character on all conjugacy classes of the point group $G_{\text{sym}}$. Its runtime is therefore

$$\mathcal{O}(|L_k| \cdot C_G) = \mathcal{O}(k \cdot C_G),$$

where $C_G$ is the number of conjugacy classes (a small constant for crystallographic point groups), with again $\mathcal{O}(k)$ memory. When an explicit invariant basis is required, we form the projectors

$$\Pi_\ell = \frac{1}{|G_{\text{sym}}|} \sum_{g \in G_{\text{sym}}} \rho^{(\ell)}(g)$$

on each SO(3) block, which costs $\mathcal{O}(|G_{\text{sym}}|d_\ell^2 + d_\ell^3)$ per spin, where $d_\ell = 2\ell + 1$. Summed over all $\ell \leq k$, this yields at most

$$\mathcal{O}(|G_{\text{sym}}|k^3 + k^4) \text{ time and } \mathcal{O}(k^3) \text{ memory.}$$

For the tensor orders considered in this work ($k \leq 4$), the entire basis generation is a one-off precomputation that takes negligible time and memory compared to training or inference of the neural network.

**Case Study (elastic tensor).** For the elastic tensor (rank-4), Algorithm 1 first performs an unconstrained SO(3) decomposition of $V^{\otimes 4}$ and then imposes the major/minor index symmetries of the elastic stiffness tensor. This reduces the 81-dimensional space of rank-4 tensors to the 21-dimensional elasticity space, which decomposes under SO(3) as

$$\mathcal{C} \cong 2 \times \ell = 0 \ \oplus \ 2 \times \ell = 2 \ \oplus \ 1 \times \ell = 4,$$

i.e.

$$M_0^{\text{elastic}}[0] = 2, \quad M_0^{\text{elastic}}[2] = 2, \quad M_0^{\text{elastic}}[4] = 1.$$

The corresponding dimensions satisfy

$$2 \times (2 \cdot 0 + 1) \ + \ 2 \times (2 \cdot 2 + 1) \ + \ 1 \times (2 \cdot 4 + 1) = 2 + 10 + 9 = 21,$$

matching the 21 independent components of the elastic tensor. Given a crystal point group $G_{\text{sym}}$, the procedure ApplySym then computes, for each spin $\ell$, the multiplicity $n_\ell$ of the trivial representation in the restriction of the $(2\ell + 1)$-dimensional SO(3) irrep to $G_{\text{sym}}$ via a class-sum character test. For a cubic crystal with $G_{\text{sym}} = O_h$, this yields

$$n_0 = 1, \qquad n_2 = 0, \qquad n_4 = 1,$$

and hence

$$M_{\text{final}}^{\text{elastic}}[0] = 2 \times n_0 = 2, \quad M_{\text{final}}^{\text{elastic}}[2] = 2 \times n_2 = 0, \quad M_{\text{final}}^{\text{elastic}}[4] = 1 \times n_4 = 1.$$

The total number of invariant coefficients is therefore

$$2 + 0 + 1 = 3,$$

in one-to-one correspondence with the three independent elastic constants $(C_{11}, C_{12}, C_{44})$ of a cubic crystal. For each spin $\ell$, we finally construct explicit invariant bases $\{\mathcal{B}_a^{(\ell)}\}_{a=1}^{M_{\text{final}}^{\text{elastic}}[\ell]}$ via the projector

$$\Pi_\ell = \frac{1}{|G_{\text{sym}}|} \sum_{g \in G_{\text{sym}}} \rho^{(\ell)}(g).$$

The network predicts only the scalar coefficients of these irreducible basis tensors, so any reconstructed elastic tensor is guaranteed to satisfy both the index symmetries and the crystal point-group symmetry by construction.

# H    Guarantees for Symmetry and Accuracy

Building on the unconstrained decomposition and symmetry compatibility step in Section 3.1, we fix a canonical irreducible basis for the symmetry constraint G and learn only the associated basis coefficients. Reconstruction is restricted to the symmetry-compatible subspace $S_G$. Operationally, the group projector $P_G$ implements the constraint by zeroing symmetry-forbidden coefficients and enforcing equality among symmetry-related ones, yielding an exactly constrained, compact parameterization. The number of free degrees of freedom drops from the ambient dimension D to the symmetry dimension $d_G$; estimation noise may perturb coefficient magnitudes but cannot introduce symmetry-forbidden patterns, so symmetry is guaranteed by construction.

Below we make this precise. Theorem 1 characterizes $P_G$ via character projectors and explains why the reconstruction is exactly symmetry compliant. Proposition 3 shows that projecting cannot increase the per-sample Frobenius error when the ground truth lies in $S_G$. Theorem 2 gives the expected risk identity $\mathbb{E}\|P_G(Y) - T^\star\|_F^2 = \mathrm{tr}(P_G\Sigma)$ and reduces to $\sigma^2 d_G$ under isotropic noise, while Corollary 1 shows that stronger symmetry implies no larger risk and typically a smaller one. Together these results justify the claim from the introduction that predicting symmetry-allowed irreducible components preserves symmetry exactly and improves accuracy in expectation.

**Setting**    Let $(\mathcal{V}, \langle \cdot, \cdot \rangle)$ be the real inner-product space of order-$k$ tensors with Frobenius norm $\| \cdot \|_F$ and dimension $D = \dim \mathcal{V}$. Let $G$ be the crystallographic point group acting orthogonally via $\rho : G \to O(\mathcal{V})$. By complete reducibility, $\mathcal{V} = \bigoplus_\ell \mathcal{V}^{(\ell)}$, where $\mathcal{V}^{(\ell)}$ is the isotypic component for irrep $\ell$ with multiplicity $m_\ell$ and irrep dimension $r_\ell$, so $\dim \mathcal{V}^{(\ell)} = m_\ell r_\ell$. Given the set of *allowed* irreps $\mathcal{L}_G$, define the symmetry subspace

$$S_G = \bigoplus_{\ell \in \mathcal{L}_G} \mathcal{V}^{(\ell)}, \qquad d_G = \dim S_G,$$

and let $P_G : \mathcal{V} \to S_G$ be the orthogonal projector.

**Theorem 1** (Character projectors and exact symmetry). *Define*

$$\Pi_\ell = \frac{r_\ell}{|G|} \sum_{g \in G} \overline{\chi_\ell(g)}\, \rho(g), \qquad P_G = \sum_{\ell \in \mathcal{L}_G} \Pi_\ell.$$

Then $\Pi_\ell$ projects onto $\mathcal{V}^{(\ell)}$ and $P_G$ projects onto $S_G$. Consequently $P_G$ zeros all forbidden components and ties symmetry-equivalent ones, yielding an exactly symmetry-compliant reconstruction.

**Theorem 2** (Risk identity and isotropic case). *Let $\mathbb{E}[\varepsilon] = 0$ and $\Sigma = \mathbb{E}[\varepsilon \varepsilon^\top] \succeq 0$.*

Then

$$\mathbb{E}\|P_G(Y) - T^\star\|_F^2 = \mathrm{tr}(P_G\Sigma), \qquad \mathbb{E}\|Y - T^\star\|_F^2 = \mathrm{tr}(\Sigma).$$

If the noise is isotropic, $\Sigma = \sigma^2 I$, then

$$\mathbb{E}\|P_G(Y) - T^\star\|_F^2 = \sigma^2 d_G, \qquad \mathbb{E}\|Y - T^\star\|_F^2 = \sigma^2 D.$$

**Proposition 3**    Inequality of point projection. If $T^\star \in S_G$ and $Y = T^\star + \varepsilon$, then

$$\|P_G(Y) - T^\star\|_F \leq \|Y - T^\star\|_F,$$

with equality if and only if $\varepsilon \in S_G$.

**Corollary 1**    Monotonicity under stricter symmetry. If $H$ is stricter than $G$ so that $S_H \subset S_G$ and $P_H \preceq P_G$, then for any $\Sigma \succeq 0$,

$$\mathrm{tr}(P_H\Sigma) \leq \mathrm{tr}(P_G\Sigma) \leq \mathrm{tr}(\Sigma).$$

In particular, for $\Sigma = \sigma^2 I$, $\sigma^2 d_H \leq \sigma^2 d_G \leq \sigma^2 D$.

**Finite-sample implication.** For linear heads mapping features into $S_G$ with Frobenius norm bounded by $B$, the empirical Rademacher complexity scales as

$$\mathfrak{R}_n \ \leq \ \frac{B}{\sqrt{n}}\sqrt{d_G},$$

so restricting to $S_G$ reduces capacity from a $\sqrt{D}$ to a $\sqrt{d_G}$ factor, improving sample efficiency.

In conclusion, Predicting irreducible components confines outputs to $S_G$. This enforces symmetry by construction and reduces expected error from $\mathrm{tr}(\Sigma)$ to $\mathrm{tr}(P_G\Sigma)$, which equals $\sigma^2 d_G$ under isotropic noise. As symmetry becomes stricter and $d_G$ shrinks, accuracy improves in expectation.

## I   LARGE LANGUAGE MODEL USAGE

We used a large language model (ChatGPT) solely for language polishing—i.e., grammar, wording, and minor stylistic edits to author-written text. The LLM was not used for research ideation, method or experiment design, data analysis, drafting substantive technical content, generating equations, or producing/altering references. All ideas, models, proofs, experiments, figures, and conclusions are the authors' own, and any edited text was reviewed and approved by the authors.

## J   THE SENSITIVITY DISCUSSIONS TO $\ell_{max}$ AND CHANNELS

**Choice of $\ell_{\mathbf{max}}$.** Our architecture is implemented using the SO(2) reformulation of SO(3) convolutions (eSCN), while the feature space is still organized as SO(3) irreps up to degree $\ell_{\max}$. Thus $\ell_{\max} = \ell$ plays the usual role of the maximum angular momentum and controls the angular resolution of the model.

From representation theory, the *target tensor* itself only needs irreps up to a certain order. Concretely, symmetric rank-2 tensors: $\ell \leq 2$, pair-symmetric rank-3 tensors: $\ell \leq 3$, elastic rank-4 tensors: $\ell \leq 4$. In principle, this means that a model predicting only a symmetric rank-2 property could already be implemented with $\ell_{\max} = 2$.

In our experiments, we choose $\ell_{\max}$ slightly more conservatively for two reasons.

First, for dielectric and piezoelectric prediction, we set $\ell_{\max} = 3$. For piezoelectric tensors (pair-symmetric rank-3), $\ell_{\max} = 3$ matches the maximal physically allowed $\ell$. For dielectric tensors (symmetric rank-2), $\ell_{\max} = 3$ is one order higher than strictly required ($\ell \leq 2$ would be sufficient), but it allows us to use a unified architecture across the second- and third-order tasks and provides slightly richer intermediate angular features. In all cases, the final tensor head only uses the irreps that are consistent with the target tensor's index symmetries.

Second, for elastic tensors, we set $\ell_{\max} = 4$, which fully covers the irreps needed for rank-4 elasticity while avoiding the cubic growth in cost with $\ell_{\max}$ inherent to eSCN-like convolutions.

These settings are also consistent with the hyperparameter choices reported in GMTNet. In Table 5, we further provide an ablation over $\ell_{\max}$, showing that smaller values (e.g., $\ell_{\max} = 2$ for dielectric, $\ell_{\max} = 2, 3$ for piezoelectric) do not lead to significant gains and that our default choice offers a good trade-off between accuracy and computational cost.

**Degree-wise channels.** Channel counts are then tuned under a fixed compute budget: we allocate more channels to low and mid degrees ($\ell$=0, 1, 2), which carry most of the physical signal, and fewer channels to higher degrees. This strategy provides sufficient capacity to model complex high-order anisotropy while maintaining stable training and reasonable runtime. For each degree $\ell$ and parity (even/odd), we assign channels in a monotonically decreasing manner with $\ell$, roughly following a geometric decay (e.g., 32, 16, 8, 4, 2 for $\ell = 0,1,2,3,4$). Low and mid degrees ($\ell = 0,1,2$) are given significantly more channels because they carry most of the physical signal (scalar, vector, and low-order anisotropy), while higher degrees ($\ell \geq 3$) are assigned fewer channels and mainly serve to capture fine-grained anisotropy. Even and odd irreps at the same $\ell$ share the same width, reflecting that we do

Table 5: $\ell_{\max}$ sensitivity experiments on the JARVIS-DFT dataset.

| Dataset | $\ell_{\max}$ | Fnorm ($\downarrow$) | EwT 25% ($\uparrow$) | EwT 10% ($\uparrow$) | EwT 5% ($\uparrow$) | Total Time (s) ($\downarrow$) |
|---|---|---|---|---|---|---|
| Dielectric | 2 | 3.31 | **85.9%** | 52.1% | **29.9%** | **711** |
| Dielectric | 3 | **3.23** | 85.8% | **53.2%** | 29.8% | 798 |
| Piezoelectric | 3 | **0.29** | 49.3% | 49.3% | **49.2%** | **288** |
| Piezoelectric | 4 | 0.30 | 49.1% | **48.9%** | 48.5% | 314 |
| Piezoelectric | 5 | 0.33 | 49.2% | **48.9%** | 48.6%% | 454 |
| Elastic | 4 | 64.23 | **66.5%** | 26.5% | **12.1%** | **14121** |
| Elastic | 5 | 64.61 | 65.7% | 26.1% | **12.1%** | 16237 |
| Elastic | 6 | **63.88** | **66.5%** | **26.9%** | 12.0% | 18711 |

not impose any prior bias between parities. This degree-wise allocation gives a good trade-off between expressivity of high-order components and computational efficiency.

To assess the sensitivity of this design, we perform a channel-allocation ablation under a comparable total width (Table 6). Besides our default monotonically decreasing allocation (e.g., $32 \times 0e/0o, 16 \times 1e/1o, 8 \times 2e/2o, 4 \times 3e/3o, 2 \times 4e/4o$), we consider two alternatives: (i) an approximately flat allocation with 12 channels per parity at each degree (e.g., $12 \times 0e/0o, \ldots, 12 \times 4e/4o$), where the value 12 is chosen so that the overall number of channels is similar to the default setting, and (ii) a high-degree-heavy allocation that reverses the pattern and allocates most channels to $\ell = 3, 4$. Across all three tensor prediction tasks, our default scheme achieves the best or near-best Fnorm and EwT, while the high-degree-heavy configuration clearly degrades accuracy (e.g., for elastic tensors, Fnorm increases from 64.23 to 69.77 and EwT 25% drops from 66.5% to 61.5%). The flat allocation is only slightly worse than the default, indicating that the model is reasonably robust to moderate changes in the exact per-degree counts. Overall, these results support our design principle: allocating more capacity to low and mid degrees ($\ell = 0, 1, 2$) is important for performance, whereas over-emphasizing high degrees is not beneficial and mainly increases computational cost.

Table 6: channels sensitivity experiments on the JARVIS-DFT dataset.

| Dataset | hidden irreps | Fnorm ($\downarrow$) | EwT 25% ($\uparrow$) | EwT 10% ($\uparrow$) | EwT 5% ($\uparrow$) |
|---|---|---|---|---|---|
| Dielectric | 32x0e+32x0o+16x1e+16x1o+8x2e+8x2o+4x3e+4x3o | **3.23** | **85.8%** | **53.2%** | **29.8%** |
| Dielectric | 12x0e+12x0o+12x1e+12x1o+12x2e+12x2o+12x3e+12x3o | 3.43 | 83.8% | 51.2% | 26.7% |
| Dielectric | 12x0e+12x0o+12x1e+12x1o+12x2e+12x2o+12x3e+12x3o | 3.86 | 78.3% | 40.9% | 20.4% |
| Piezoelectric | 32x0e+32x0o+16x1e+16x1o+8x2e+8x2o+4x3e+4x3o | **0.29** | **49.3%** | **49.3%** | **49.2%** |
| Piezoelectric | 12x0e+12x0o+12x1e+12x1o+12x2e+12x2o+12x3e+12x3o | 0.33 | 48.9% | 48.7% | 48.4% |
| Piezoelectric | 12x0e+12x0o+12x1e+12x1o+12x2e+12x2o+12x3e+12x3o | 0.41 | 47.3% | 47.2% | 46.8% |
| Elastic | 32x0e+32x0o+16x1e+16x1o+8x2e+8x2o+4x3e+4x3o+2x4e+2x4o | **64.23** | 66.5% | **26.5%** | **12.1%** |
| Elastic | 12x0e+12x0o+12x1e+12x1o+12x2e+12x2o+12x3e+12x3o+12x4e+12x4o | 64.41 | **66.8%** | 23.4% | 11.5% |
| Elastic | 2x0e+2x0o+4x1e+4x1o+8x2e+8x2o+16x3e+16x3o+32x4e+32x4o | 69.77 | 61.5% | 20.9% | 7.5% |

## K    DICUSSIONS ABOUT IRREDNET AND BASELINES

**Relation to prior tensor predictors.** Our work is closely related to previous O(3)-equivariant GNNs for crystal tensor prediction, most notably GMTNet and ETGNN, which motivates a careful comparison. At a high level, all these methods represent crystals as graphs and use equivariant message passing. The main differences lie in how symmetries are enforced, how tensor properties are parameterized, and how the equivariant backbone is organized.

First, the mechanism for symmetry enforcement is mathematically distinct. GMTNet employs a projection-based approach: the network operates in a larger latent space, and an equivariant symmetry enforcement module projects features onto the symmetric subspace (via Wigner-D averaging) during the forward pass1. While this enforces symmetry during training, it implies that the network allocates capacity to feature dimensions that are subsequently masked out. Furthermore, because this projection relies on floating-point Wigner-D operations, it introduces small numerical residuals, necessitating tolerance-based

adjustments to enforce strict zero or equality constraints. In contrast, IrredNet employs basis reparameterization. We analytically determine the symmetry-adapted irreducible basis offline using group representation theory and construct the network to predict only the scalar coefficients of this basis. We do not "mask" invalid channels; rather, we structurally remove the parameters associated with them from the hypothesis space entirely. As a result, the output is guaranteed to be symmetric by construction (structural exactness), eliminating the need for numerical tolerance thresholds to handle floating-point noise, which is particularly advantageous for high-order tensors where numerical projection can become fragile.

Second, the mathematical formulation of the prediction head is fundamentally different. GMTNet follows a physics-simulation approach: it models an intermediate physical potential (e.g., strain energy or stress field) and employs automatic differentiation (e.g., torch.autograd) to derive the target tensor as a gradient (e.g., $C = \partial\sigma/\partial\epsilon$).IrredNet, on the other hand, follows a representation-theoretic approach. We do not simulate physical potentials or compute gradients. Instead, the Irreducible Decomposition Block directly predicts the scalar coefficients for the pre-computed, symmetry-adapted basis via a linear projection. The final tensor is reconstructed as a linear combination of these basis tensors. This design reduces the effective degrees of freedom and decouples the prediction from the physical response path, leading to improved data efficiency and accuracy for complex high-order tensors.

Generalizable Backbone IntegrationFinally, our equivariant backbone is designed to be property-agnostic. We build on a modern SO(3)-equivariant GNN implemented via SO(2)-steerable irreps, producing a rich irreducible feature space. Different tensor orders (dielectric, piezoelectric, elastic) are handled simply by attaching the appropriate irreducible-basis head, without redesigning perturbation schemes or response modules for each property.Together, the structural exactness of the irreducible-basis parameterization and the direct coefficient prediction allow IrredNet to provide a distinct and complementary methodology to prior projection-based approaches, offering superior stability and performance on high-order tensor benchmarks.

