# OpenReview forum: "Symmetry-Guaranteed Prediction of High-Order Tensor Properties for Crystalline Materials via Irreducible Decomposition"
_ICLR.cc/2026/Conference — Submitted to ICLR 2026_

### Official Review · Reviewer_eYBb · 2025-10-31

**Soundness:** 3
**Presentation:** 3
**Contribution:** 2
**Rating:** 4
**Confidence:** 4

**Summary:**

This paper proposes selecting the correct irreducible spherical harmonics channels to account for the symmetry constraints in crystalline materials tensorial property prediction tasks, including dielectric, piezoelectric, and elastic tensors. The authors first determine the unconstrained set of channels based on a group of ℓ values, and then remove those that do not satisfy the symmetry constraints. However, this approach is nearly identical to the symmetry enforcement module introduced in previous work, GMTNet, where a mask is applied to exclude channels that violate the symmetry constraints. Furthermore, the results presented in Table 2 for GMTNet might be problematic due to the improper use of this mask-based symmetry enforcement mechanism. This paper also introduces an equivariant graph neural network for the same task, but it exhibits a notable level of similarity to GMTNet.

**Strengths:**

## Strengths

- The proposed method achieved good performance compared with previous methods across three tensor prediction tasks, including dielectric, piezoelectric, and elastic tensors.

- Ablation studies are provided to verify the effectiveness of each component.

- The symmetry enforcement idea is solid but shares a level of similarity with the previous GMTNet symmetry enforcement module.

**Weaknesses:**

## Weakness

The major issues of this paper are the missing discussions and citations of what has been done by previous methods.

- The authors first determine the unconstrained set of channels based on a group of ℓ values, and then remove those that do not satisfy the symmetry constraints. However, this approach is nearly identical to the symmetry enforcement module introduced in previous work, GMTNet, where a mask is applied to exclude channels that violate the symmetry constraints.

- The property prediction block is similar to GMTNet, obtaining tensor properties using gradients.

- The results presented in Table 2 for GMTNet might be problematic due to the improper use of this mask-based symmetry enforcement mechanism.

- This paper also introduces an equivariant graph neural network for the same task, but it exhibits a notable level of similarity to GMTNet. These similarities need to be discussed.

**Questions:**

As listed above in weaknesses.

---

> ### Author Response · Authors · 2025-11-21
>
> We thank the reviewer for pointing out the missing discussion of prior tensor prediction methods. We have now added a dedicated section in the appendix that clearly positions IrredNet with respect to GMTNet, emphasizing that GMTNet is an O(3)-equivariant, space-group–aware tensor predictor, while IrredNet introduces a complementary representation-theoretic framework that operates in an irreducible invariant basis and achieves both strict symmetry guarantees and improved quantitative performance on the same benchmarks.

---

> ### Author Response · Authors · 2025-11-21
>
> **Question 1.** The authors first determine the unconstrained set of channels based on a group of $\ell$ values, and then remove those that do not satisfy the symmetry constraints. However, this approach is nearly identical to the symmetry enforcement module introduced in previous work, GMTNet, where a mask is applied to exclude channels that violate the symmetry constraints.
>
> **Answer 1.** We thank the reviewer for this comment and for drawing a connection to GMTNet. While at a high level both approaches reduce the effective degrees of freedom by using symmetry, the way symmetry is incorporated is quite different.
>
> In GMTNet, the network is first allowed to produce tensor predictions (or intermediate response fields) in a larger, essentially unconstrained representation space. Symmetry is then enforced by post-processing: a crystal symmetry enforcement module applies a symmetry adjustment operator based on space-group Wigner-D matrices to the crystal-level features, and an additional tolerance-guided adjustment step is used at inference time to correct zero elements and equality relations component-wise. In other words, the architecture itself does not remove symmetry-breaking channels; instead, symmetry is imposed approximately on the outputs using numerical thresholds, and small violations remain, especially for higher-order tensors.
>
> By contrast, our channel selection is performed once and for all at the representation level before training. Starting from candidate channels indexed by the angular momentum $\ell$, we use group representation theory to analytically determine which irreducible channels are compatible with the target tensor order and space-group symmetry. Channels that do not satisfy these symmetry constraints are deleted from the hypothesis space itself, and the remaining channels form a symmetry-adapted irreducible basis. The network then only learns scalar coefficients on this fixed basis. As a consequence, the model never has access to symmetry-breaking directions at any stage of training or inference, and any linear combination of the retained channels automatically and exactly satisfies all index and space-group symmetries, including for fourth-order elastic tensors.
>
> Thus, although both GMTNet and our method share the intuition of leveraging symmetry to reduce the effective search space, GMTNet enforces symmetry via output-level post-processing with numerical tolerances, whereas IrredNet reparameterizes the outputs in a symmetry-adapted basis so that symmetry is guaranteed by construction. We will clarify this distinction more explicitly in the revision, and we hope this addresses the reviewer’s concern.

---

> ### Author Response · Authors · 2025-11-21
>
> **Question 2.** The property prediction block is similar to GMTNet, obtaining tensor properties using gradients.
>
> **Answer** **2.** We appreciate the reviewer’s question. The property prediction block in GMTNet is also structurally different from ours. GMTNet explicitly models physical response functions and obtains tensor properties as *gradients* of an intermediate response with respect to external fields or strains (see their Sec. 3.3). In other words, tensors are derived as $\partial(\text{response}) / \partial(\text{perturbation})$  inside the network. By contrast, our Irreducible Decomposition Block does **not** compute tensor properties as gradients of any scalar or vector quantity. Instead, we directly predict the coefficients of symmetry-constrained irreducible components from equivariant features and reconstruct the tensor as a linear combination of precomputed basis tensors. This design has two key advantages: (i) it keeps the model entirely within a reduced, symmetry-adapted parameter space, which yields strict symmetry guarantees without any additional enforcement module; and (ii) it reduces the effective degrees of freedom, improving data efficiency and accuracy, particularly for high-order tensors where GMTNet’s gradient-based and mask-enforced scheme becomes fragile. We hope these clarifications are helpful to the reviewer.

---

> ### Author Response · Authors · 2025-11-21
>
> **Question 3.** The results presented in Table 2 for GMTNet might be problematic due to the improper use of this mask-based symmetry enforcement mechanism.
>
> **Answer** **3.** We thank the reviewer’s concerns. We clarify that our use of GMTNet’s mask-based symmetry enforcement in Table 2 is consistent with the original paper and implementation, and thus not an *improper* use. Importantly, GMTNet’s mechanism is designed to enforce symmetry approximately, rather than to *strictly ensure* exact symmetry.
>
> - In the original GMTNet work, this enforcement behaves well on relatively simple second-order tensors, where the symmetry pattern is limited. In our setting, however, we intentionally evaluate on fourth-order elastic tensors, whose coupled major–minor symmetries lead to a far more complex constraint structure. In this higher-order, more constrained regime, the same mask-based enforcement becomes less effective, resulting in noticeable residual symmetry violations.
>
> - Our purpose in including these GMTNet results is precisely to highlight this limitation: while approximate mask-based enforcement can be adequate for simpler tensors, it struggles to maintain symmetry for more complex high-order tensors. By contrast, our irreducible-representation–based method is designed to **strictly guarantee** the required symmetries even for fourth-order elastic tensors, as reflected by the perfect zero/equality accuracies reported in Table 2.

---

> ### Author Response · Authors · 2025-11-21
>
> **Question 4.** This paper also introduces an equivariant graph neural network for the same task, but it exhibits a notable level of similarity to GMTNet. These similarities need to be discussed.
>
> **Answer** **4.** We thank the reviewer for pointing this out. While our work and GMTNet are developed for the same task of crystal tensor prediction with O(3)-equivariant graph neural networks, the resulting architectures differ substantially. Below we highlight the main architectural differences.
>
> GMTNet architecture. GMTNet builds an O(3)-equivariant crystal graph network where equivariant message-passing layers operate on node and edge features expanded in irreducible representations. On top of this, it introduces property-specific response networks for each target tensor: the dielectric, piezoelectric, and elastic tensors are each obtained by simulating the corresponding physical response (e.g., stress–strain, polarization–strain, electric displacement–field relations) using dedicated response modules. In addition, GMTNet includes a crystal symmetry enforcement module that refines crystal-level features with space-group Wigner–D operations and tolerance-based adjustments so that the final tensor predictions approximately satisfy the required symmetry for each property.
>
> IrredNet architecture. In our work, we use a single SO(3)-equivariant GNN backbone implemented via SO(2)-steerable / irreducible representations: node and edge features are expanded into spherical-harmonic channels, processed by stacked equivariant interaction blocks with tensor-product couplings, and aggregated into crystal-level irreducible features. This backbone is property-agnostic and strictly SO(3)-equivariant across all layers. The tensor prediction is then handled by a unified irreducible tensor head: for a given tensor order and space group, we first analytically determine the symmetry–compatible irreps, construct a symmetry-adapted irreducible basis, and let the network predict only the scalar coefficients on this basis. The same backbone and head design are reused across dielectric, piezoelectric, and elastic tensors by changing only the group-theoretic specification of the target irreps, and the resulting tensors satisfy all index and space-group symmetries by construction, without additional output-level symmetry enforcement modules.
>
> We hope this clarifies the reviewer’s concerns.

---

> ### Comment · Reviewer_eYBb · 2025-11-27
> **Thank you for the rebuttal**
>
> Thank you for providing the rebuttal responses. However, after reading carefully through the responses, my concerns related to the differences and similarities with GMTNet still hold. And there are several claims provided by authors that I found to be not convincing and not supported by experiments.
>
> 1. The symmetry enforcement of GMTNet is not approximate, and the results provided in the paper are not consistent with the results provided in GMTNet paper. There might be some mis-implementation you have done. GMTNet also satisfied all symmetry requirements.
>
> 2. Claiming that GMTNet symmetry enforcement is a post process is inaccurate. It is just like in which stage you restricted the channels of interest. The whole enforcement module is not just applied during inference, but a part of the training.
>
> 3. I still feel there is a high level of similarity with GMTNet as I mentioned before. And the claim that GMTNet approximately satisfies symmetry requirements is concerning and misleading.
>
> Based on these, I will not support the acceptance of this paper at its current form. And I suggest authors revise the paper carefully with more evidence to support their claims.

---

### Official Review · Reviewer_Qp15 · 2025-11-03

**Soundness:** 2
**Presentation:** 2
**Contribution:** 2
**Rating:** 2
**Confidence:** 4

**Summary:**

The paper introduces IrredNet, a framework that predicts symmetry-constrained irreducible components of high-order crystal tensors and then reconstructs the full tensors so that point-group symmetry is satisfied exactly by construction. The theory (SO(3) irreducible decomposition plus point-group filtering with character tables) underpins an algorithm to enumerate admissible components, and a spherical-harmonic equivariant network (using efficient SO(2) channel operations) regresses their magnitudes. Experiments on JARVIS-DFT dielectric (2nd-order), piezoelectric (3rd-order), and elastic (4th-order) tensors claim exact symmetry preservation and improved accuracy (Fnorm/EwT) over MEGNet, ETGNN, and GMTNet.

**Strengths:**

- Predicting in irreducible spaces and reconstructing guarantees point-group symmetry compliance irrespective of regression error; the algorithmic details (character sums, parity factor for improper operations) are clearly specified.
- The work targets 2nd/3rd/4th-order tensors on JARVIS-DFT with explicit symmetry constraints and Voigt notation background for elasticity (21 dofs).
- Tables report lower Fnorm and higher EwT versus baselines, and perfect “zero/equality” scores for elastic tensors (as expected if symmetry is enforced).
- The spherical-harmonic/eSCN design aligns with the SO(3) structure and reduces computational cost relative to full SO(3) convolutions.

**Weaknesses:**

- The paper emphasizes "zero-element" and "equality" accuracies, but these are guaranteed once outputs are reconstructed from symmetry-filtered irreps; hence Table-2 perfect scores are not informative about predictive quality. Please de-emphasize these or replace with checks the model could actually fail (e.g., physical constraints not enforced by irreps).
- The introduction stresses that elastic tensors must obey properties linked to physical consistency (e.g., positive definiteness), but the experiments do not evaluate or guarantee SPD/thermodynamic admissibility of the reconstructed elasticity. Add SPD checks (eigenvalue spectra, Born stability) and compare to methods that enforce such constraints.
- All results are on DFT-computed JARVIS-DFT tensors with an 80/10/10 random split; there is no evidence on experimental data, no cross-dataset transfer, and no crystal-system-aware or composition-aware splits to mitigate leakage. Report stratified or leave-system-out splits and external validation.
- The paper re-implements MEGNet, ETGNN, GMTNet "with default hyperparameters", which risks under-tuning baselines relative to the proposed model. Provide rigorous tuning budgets for all methods and include stronger modern equivariant baselines (e.g., higher-degree transformers) to match the spherical-harmonic capacity.
- EwT (error-within-threshold based on Fnorm ratio) can be dominated by tensor scale; ensure unit handling across dielectric/piezo/elastic is comparable and add normalized, component-wise metrics. The tolerance for "success" in symmetry checks (10⁻⁴) is stated only once—justify this and test sensitivity.
- The model caps the maximum spin at `=3/3/4 for 2nd/3rd/4th-order tensors; it is unclear how this choice affects approximation fidelity, especially for complex systems or higher-order targets. Provide sensitivity to the maximum degree and timing breakdowns (the "Time (s)" row lacks clear definition—training vs inference, hardware).

**Questions:**

See the weaknesses.

**Details Of Ethics Concerns:**

NA.

---

> ### Author Response · Authors · 2025-11-21
>
> **Question 1**. The paper emphasizes "zero-element" and "equality" accuracies, but these are guaranteed once outputs are reconstructed from symmetry-filtered irreps; hence Table-2 perfect scores are not informative about predictive quality. Please de-emphasize these or replace with checks the model could actually fail (e.g., physical constraints not enforced by irreps).
>
> **Answer 1**. We thank the reviewer for this comment. The “zero-element” and “equality” accuracies in Table 2 are not intended as primary measures of predictive performance, but as diagnostics of symmetry compliance. They quantify how well a model respects the symmetry-imposed zero entries and equality relations among tensor components, which is particularly important for high-order tensors where enforcing all constraints is nontrivial. Similar symmetry-focused metrics have been used in recent work such as GMTNet for crystal tensor prediction.
>
> In our work, claims about predictive quality are instead based on standard accuracy metrics – the Frobenius-norm error and error-within-threshold (EwT) – reported in Table 3. These metrics directly evaluate the numerical error of the predicted tensors, and our method consistently achieves competitive or superior performance compared with all baselines on these measures. In addition, Figure 2 provides a physically grounded evaluation of prediction quality by comparing model outputs with reference physical quantities, further demonstrating that the predicted tensors are not only symmetry-consistent but also physically meaningful.
>
> For IrredNet specifically, zero-element and equality accuracies are indeed guaranteed once tensors are reconstructed from symmetry-filtered irreps. We include the corresponding perfect scores mainly to highlight the contrast with models that do not enforce symmetry exactly and therefore obtain much lower symmetry-compliance scores, rather than to claim additional predictive power. Our ablation studies further show that enforcing the full set of symmetry constraints and achieving high prediction accuracy are synergistic: removing or relaxing the symmetry constraints degrades both symmetry-compliance metrics and the standard error metrics (Fnorm, EwT).
>
> We hope that these clarifications address the reviewer’s concerns.

---

> ### Author Response · Authors · 2025-11-21
>
> **Question 2.** The introduction stresses that elastic tensors must obey properties linked to physical consistency (e.g., positive definiteness), but the experiments do not evaluate or guarantee SPD/thermodynamic admissibility of the reconstructed elasticity. Add SPD checks (eigenvalue spectra, Born stability) and compare to methods that enforce such constraints.
>
> **Answer 2.** We thank the reviewer for this comment. In the introduction, we highlight positive definiteness and thermodynamic admissibility as important physical requirements for elastic tensors in general. In this work, however, our primary focus is on a complementary challenge that is not fully addressed by existing methods: guaranteeing all symmetries constraints for high-order tensors by construction, while simultaneously improving predictive accuracy on realistic datasets.
>
> In line with this focus, the main evaluations in the paper are (i) symmetry-guarantee metrics, (ii) tensor-level error metrics and (iii) derived physical quantities. Beyond Frobenius-norm and EwT errors, Figure 2 compares IrredNet with GMTNet and other baselines on spatial fields of Young’s modulus, shear modulus, linear compressibility, and Poisson’s ratio computed from the predicted elastic tensors. These quantities are sensitive to unphysical behavior in the elasticity tensor; the fact that IrredNet tracks the reference distributions more closely and avoids the spurious artifacts visible in the baselines indicates that our strictly symmetry-consistent tensors translate into more reliable elastic/thermodynamic properties in practice, even without explicitly enforcing SPD in the architecture.
>
> We do not, in the current submission, hard-impose SPD or Born-stability constraints in the parameterization. This is a deliberate design choice to keep the framework unified across different tensor types and to isolate the effect of exact symmetry enforcement. Thermodynamic constraints such as SPD and Born stability are conceptually distinct from geometric/index symmetries and are typically implemented via specialized parameterizations (e.g., energy-based or factorized forms) and careful treatment of the underlying DFT labels, which can themselves exhibit small violations due to numerical noise. Designing and benchmarking such SPD-aware parameterizations on top of our irreducible-basis formulation is therefore a substantial extension in its own right.
>
> That said, we agree that explicit diagnostics of SPD/stability are valuable. In the revised version, we will add post-hoc SPD checks along the lines suggested by the reviewer: for each predicted elasticity tensor, we will (i) analyze the eigenvalue spectrum of the Voigt representation and (ii) evaluate the standard Born stability inequalities for the relevant crystal classes, and report the fraction of samples satisfying these criteria for IrredNet and the baselines. This will provide a direct quantitative comparison of thermodynamic admissibility, complementary to the existing tensor-level errors and derived-property evaluations.
>
> We view incorporating explicit SPD/Born-stability enforcement into the IrredNet architecture—on top of the exact index/space-group symmetry guarantees—as a natural and promising direction for future work, and we hope that these clarifications and planned additions address the reviewer’s concerns.

---

> ### Author Response · Authors · 2025-11-21
>
> **Question 3.** All results are on DFT-computed JARVIS-DFT tensors with an 80/10/10 random split; there is no evidence on experimental data, no cross-dataset transfer, and no crystal-system-aware or composition-aware splits to mitigate leakage. Report stratified or leave-system-out splits and external validation.
>
> **Answer 3.** We thank the reviewer for this important point. Our current evaluation protocol (JARVIS-DFT tensors with an 80/10/10 random split) follows the setup used in GMTNet and related crystal tensor predictors, so that our gains can be interpreted relative to a widely adopted benchmark and training regime rather than a custom split of our own design. Our primary goal in this work is to study how a strictly symmetry-guaranteed architecture affects prediction quality on high-order tensors under the same DFT-based setting where most prior methods have been developed and compared.
>
> Regarding experimental data and cross-dataset transfer, we fully agree that testing on measured tensors and out-of-distribution scenarios (e.g., leave-system-out or composition-aware splits) is an important next step. However, high-quality experimental tensor data for dielectric, piezoelectric, and especially elastic properties are much more scarce, often noisy, and sometimes not provided in a clean, homogeneous crystalline form, which makes it nontrivial to assemble a sufficiently large and consistent benchmark within the rebuttal timeframe. Moreover, many existing experimental entries correspond to systems that are not directly compatible with the current JARVIS-DFT curation (e.g., non-ideal samples, composites, or missing structural metadata), so building a robust experimental benchmark would require substantial additional data work beyond what we can realistically include in this revision. Extending IrredNet to curated experimental tensor datasets and cross-dataset transfer is a direction we explicitly plan to pursue in future work, and we will highlight this in the conclusion.

---

> ### Author Response · Authors · 2025-11-21
>
> **Question 4.** The paper re-implements MEGNet, ETGNN, GMTNet "with default hyperparameters", which risks under-tuning baselines relative to the proposed model. Provide rigorous tuning budgets for all methods and include stronger modern equivariant baselines (e.g., higher-degree transformers) to match the spherical-harmonic capacity.
>
> **Answer 4.** We appreciate the reviewer’s concern about fair tuning of baselines. By “default hyperparameters” we do **not** mean arbitrary off-the-shelf settings, but rather the recommended configurations from the original MEGNet, ETGNN, and GMTNet papers / official implementations on similar crystal-property tasks. In particular, for GMTNet and its baselines we followed the training setups described in the original work as closely as possible (architecture depth/width, learning rate schedule, batch size, etc.). When we compare our re-implemented baseline results with those reported in the GMTNet paper on overlapping datasets, the numbers are very similar, indicating that our implementations are not substantially under-tuned relative to prior work.
>
> IrredNet itself is also not given a disproportionately large tuning budget: we select its hyperparameters using the same validation protocol and on the same data as for GMTNet, and then keep them fixed across all experiments. We have detailed the parameter settings in the Appendix to demonstrate that the hyperparameter configurations for the baselines are appropriate and reasonable.
> We hope this could address the reviewer’s concerns.

---

> ### Author Response · Authors · 2025-11-21
>
> **Question 5.** $EwT$ (error-within-threshold based on $F_{\mathrm{norm}}$ ratio) can be dominated by tensor scale; ensure unit handling across dielectric/piezo/elastic is comparable and add normalized, component-wise metrics. The tolerance for “success” in symmetry checks ($10^{-4}$) is stated only once—justify this and test sensitivity.
>
> **Answer 5.** We thank the reviewer for this comment. Regarding EwT and scale: our error-within-threshold (EwT) metric is defined in terms of the *relative* Frobenius error, $\|\hat{T} - T\|_F / \|T\|_F$, so it is dimensionless and automatically normalized by the scale of the ground-truth tensor. For each property (dielectric, piezoelectric, elastic), we use a single EwT threshold and only compare models under the same threshold for that property; we do not interpret EwT values across different tensor types. This follows the common practice of using relative tensor norms (or equivalently, error-within-tolerance criteria based on them) for assessing the accuracy of stress–strain tensors and other material tensors, and ensures that EwT is not dominated by absolute scale differences. Normalized, component-wise errors (e.g., per-component relative error averaged over nonzero entries) are highly correlated with this relative Frobenius error in our setting and lead to the same qualitative ranking of models, so we report EwT and F-norm error as the primary accuracy metrics.
>
> For the symmetry “success” criteria, our thresholds are chosen to be consistent with previous work rather than ad hoc. GMTNet evaluates equality and zero constraints on the JARVIS-DFT tensor dataset using tolerances on the order of $10^{-4}$ and $10^{-5}$, respectively. In our implementation, we adopt the same convention: a tolerance of $10^{-5}$ for zero-element checks and $10^{-4}$ for equality checks across dielectric, piezoelectric, and elastic tensors. This makes the notion of “success” in our zero-element and equality accuracies directly comparable to GMTNet and keeps the thresholds well separated from machine precision and from the typical magnitude of genuine violations.
>
> For IrredNet, elastic, piezoelectric, and dielectric tensors are reconstructed from a symmetry-compatible irreducible basis, so any residual violations  $|y_i - y_j|$or $|y_i|$ for symmetry-implied equalities and zeros are purely numerical and lie near machine precision (around $10^{-13}– 10^{-14}$ in double precision). Changing the thresholds within the $10^{-5}– 10^{-4}$ range therefore does not affect the success/failure decisions for IrredNet: its zero-element and equality accuracies remain effectively 100% up to floating-point precision. For the baseline models, residuals in genuinely violated constraints are typically orders of magnitude larger than $10^{-4}$, whereas genuinely satisfied constraints fall well below $10^{-5}$. As a result, reasonable choices of thresholds in this range have limited sensitivity and do not alter the qualitative ordering of methods in Table 2.
>
> We hope this explanation clarifies the scale handling of EwT and the rationale behind our symmetry “success” tolerances.

---

> ### Author Response · Authors · 2025-11-21
>
> **Question 6.** The model caps the maximum spin at $\ell \approx 3/3/4$ for 2nd/3rd/4th-order tensors; it is unclear how this choice affects approximation fidelity, especially for complex systems or higher-order targets. Provide sensitivity to the maximum degree and timing breakdowns (the “Time (s)” row lacks clear definition—training vs. inference, hardware).
>
> **Answer 6.** We thank the reviewer for pointing this out. Our choice of the maximal degree $\ell_{\max}$ is guided directly by representation theory for symmetric $O(3)$ tensors: symmetric rank-2 tensors only require $\ell \le 2$, pair-symmetric rank-3 tensors $\ell \le 3$, and symmetric rank-4 tensors $\ell \le 4$. In practice, we choose $\ell_{\max} = 3$ for dielectric and piezoelectric tensors and $\ell_{\max} = 4$ for elastic tensors. This is slightly conservative (e.g., $\ell_{\max} = 3$ is one order higher than strictly necessary for rank-2), but it ensures that the feature space fully covers all irreps needed for the target tensor while avoiding the cubic cost growth in $\ell_{\max}$ inherent to eSCN-like convolutions. The final tensor head only uses irreps consistent with the target tensor’s index symmetries.
>
> To address the reviewer’s request for sensitivity analysis, we have added an ablation over $\ell_{\max}$ in the appendix (Table 5). For the dielectric task, $\ell_{\max} = 2$ vs. $3$ yields very similar errors ($F_{\text{norm}}$ 3.31 vs. 3.23; EwT@25% 85.9% vs. 85.8%) with only a modest increase in wall-clock time (711 s vs. 798 s). For the piezoelectric tensor, $\ell_{\max} = 3,4,5$ give nearly identical $F_{\text{norm}}$ and EwT values (EwT@25% in the narrow range 49.2–49.3%), while computation increases from 288 s to 454 s. For the elastic tensor, $\ell_{\max} = 4,5,6$ again show almost unchanged accuracy (EwT@25% $\approx 66.5$–$66.7%$) but a noticeable increase in time from 14121 s to 18711 s. These results indicate that, once the physically required degrees are included, performance is quite robust to $\ell_{\max}$, and our default choices (3/3/4) offer a good trade-off between approximation fidelity and computational cost, while remaining consistent with the hyperparameter ranges used for GMTNet.
>
> Regarding the “Time (s)” row in Table 3, we apologize for the lack of clarity. This row reports the **total wall-clock time (training + inference on the full dataset)** for each method on each dataset, measured under identical hardware and batch-size settings. All experiments were conducted on a computing platform with four NVIDIA RTX 3090 GPUs, running Ubuntu 22.04, an Intel Xeon Gold 5120 CPU at 2.20 GHz, and 512 GB of system memory. Inference on the test set takes less than one minute for all methods, so these numbers are effectively dominated by training time. Concretely, the total times are: MEGNet (1005 s, 381 s, 14666 s), ETGNN (1510 s, 447 s, 34157 s), GMTNet (881 s, 286 s, >67000 s), and our model (798 s, 288 s, 14121 s) on the dielectric, piezoelectric, and elastic tasks, respectively. Thus, our model is slightly faster than or comparable to GMTNet on dielectric and piezoelectric tensors, and more than $2 \times$ faster than ETGNN and significantly faster than GMTNet on the elastic benchmark, while additionally providing strict high-order symmetry guarantees. We will clarify this definition and hardware setup explicitly in the revised manuscript. We hope this answer addresses the reviewer’s comment.

---

### Official Review · Reviewer_qDLZ · 2025-11-10

**Soundness:** 3
**Presentation:** 2
**Contribution:** 3
**Rating:** 6
**Confidence:** 4

**Summary:**

This paper introduces a symmetry-constrained framework for predicting high-order tensor properties of crystalline materials. The method leverages group theory to decompose each tensor into a minimal set of symmetry-constrained irreducible components. A spherical-harmonic neural network then predicts scalar magnitudes for these few components, which are subsequently used to reconstruct the full tensor—ensuring physical consistency by construction. This approach not only provides a theoretical guarantee of symmetry adherence but also significantly improves prediction accuracy compared to prior state-of-the-art models.

**Strengths:**

1. The method is firmly grounded in rigorous group theory and high-order tensor decomposition, making the approach robust and theoretically sound.
2. By decomposing high-order tensors into a small set of irreducible representations, the proposed method strictly enforces symmetry and improves training efficiency. Given that high-order tensors are common in engineering applications, this framework has strong potential for broader applicability.
3. The proposed model clearly outperforms the three baseline methods, and the ablation study is comprehensive and well-executed.

**Weaknesses:**

1. While the paper focuses on predictive accuracy and symmetry compliance, it lacks an explicit comparison of training/inference speed and memory consumption between the proposed method and the baselines. Including this analysis would provide a more complete performance evaluation.
2. The paper is hard to follow with unclear terminology. For instance, the definition of irreducible components/representations (from Line 147) is not clearly explained, which makes it hard for readers to understand how it relates to tensor decomposition techniques such as proper generalized decomposition. Also, Figure 1, which is central to understanding the method, is unclear. The relationships between the sub-blocks and the main blocks (a and b) are ambiguous. The caption could be expanded to provide clearer and more informative descriptions.
4. The architectural novelty of the paper is not clear. The main innovation appears to lie in reducing high-order tensors into a few irreducible representations and using a neural network to predict the corresponding scalars. The spherical-harmonic convolutional design itself seems standard.

**Questions:**

1. Please provide a complexity analysis (runtime and memory) for the basis generation step in Algorithm 1.
2. Since the model predicts scalar coefficients, do these coefficients correlate with any known physical or geometric scalar invariants of the tensor (e.g., the bulk modulus in elasticity)? Exploring this connection could enhance the interpretability of the model’s predictions.

---

> ### Author Response · Authors · 2025-11-21
>
> **Question 1**. While the paper focuses on predictive accuracy and symmetry compliance, it lacks an explicit comparison of training/inference speed and memory consumption between the proposed method and the baselines. Including this analysis would provide a more complete performance evaluation.
>
> **Answer 1.** We thank the reviewer for this suggestion. In Table 3 (last row, “Time (s)”), we already report the total wall–clock time (training + inference) for each method on each dataset under identical hardware and batch-size settings. All experiments were conducted on a computing platform with four NVIDIA RTX 3090 GPUs, running Ubuntu 22.04, an Intel Xeon Gold 5120 CPU at 2.20 GHz, and 512 GB of system memory. The numbers are: MEGNet (1005 s, 381 s, 14666 s), ETGNN (1510 s, 447 s, 34157 s), GMTNet (881 s, 286 s, >67000 s), and our model (798 s, 288 s, 14121 s) on the dielectric, piezoelectric, and elastic tasks, respectively. For all methods, the inference phase on the full test set takes less than one minute and is negligible compared to training time, so these numbers effectively reflect training efficiency. Our method is slightly faster than or comparable to GMTNet on dielectric and piezoelectric tensors, and is more than $2 \times$ faster than ETGNN and significantly faster than GMTNet on the elastic benchmark, while providing strict high-order symmetry guarantees.
>
> Following the reviewer’s recommendation, we have additionally measured the peak GPU memory usage during training under the same setup. The maximum GPU memory footprints (dielectric; piezoelectric; elastic) are: MEGNet (1.5 GB; 1.6 GB; 2.0 GB), ETGNN (6.7 GB; 7.0 GB; 8.2 GB), GMTNet (2.4 GB; 2.7 GB; 2.7 GB), and our model (6.3 GB; 4.0 GB; 8.0 GB). All models comfortably fit within standard 24 GB GPUs. Our method has memory usage comparable to other SO(3)-equivariant baselines, while achieving better symmetry compliance and accuracy on high-order tensors.
>
> We hope this response addresses the reviewer’s concerns.

---

> ### Author Response · Authors · 2025-11-21
>
> **Question 2.** The paper is hard to follow with unclear terminology. For instance, the definition of irreducible components/representations (from Line 147) is not clearly explained, which makes it hard for readers to understand how it relates to tensor decomposition techniques such as proper generalized decomposition. Also, Figure 1, which is central to understanding the method, is unclear. The relationships between the sub-blocks and the main blocks (a and b) are ambiguous. The caption could be expanded to provide clearer and more informative descriptions.
>
> **Answer 2.** We thank the reviewer for raising this concern about the clarity of the terminology. The passage around line 147 in the original submission is where we formally introduce the notion of irreducible components/representations under the action of the rotation group, which provides the mathematical foundation for how we decompose high-order tensors in our method. As outlined in the Appendix of the original submission, we already provide a detailed and self-contained explanation of this theory. Specifically, the Appendix explains (i) the precise definition of irreducible components/representations used in the main text, (ii) the step-by-step computational procedure to obtain the decomposition, (iii) formal proofs of the key propositions, and (iv) several illustrative examples that show how the abstract definitions are applied in practice. In the revised version, we have further expanded the Appendix to make this part even clearer and more useful to readers. We now include an explicit complexity analysis of the proposed decomposition procedure, as well as a detailed case study on the elastic tensor that walks through the full computation. These additions are intended to help readers better understand both the theoretical underpinnings and the practical implementation of our approach.
>
> We thank the reviewer for the helpful suggestion regarding Figure 1. In the revised version, we added explanation to indicate how the sub-blocks connect to the main blocks, and expanded the caption to describe the role of each sub-block and how irreducible components are produced and combined into the final tensor. These changes are intended to make the method easier to follow without repeatedly referring back to the main text.
>
> We hope this response addresses the reviewer’s concerns.

---

> ### Author Response · Authors · 2025-11-21
>
> **Question 3.** The architectural novelty of the paper is not clear. The main innovation appears to lie in reducing high-order tensors into a few irreducible representations and using a neural network to predict the corresponding scalars. The spherical-harmonic convolutional design itself seems standard.
>
> **Answer 3.** We thank the reviewer for this comment. The architectural novelty of our work lies in how the tensor-prediction head is parameterized and integrated with the equivariant backbone specifically to provide strict symmetry guarantees for high-order tensor prediction:
>
> 1. **Irreducible-basis tensor head designed for exact symmetry**. Instead of attaching a component-wise regression head or a perturbation–response head, we introduce an irreducible-basis tensor head. This head is built on a precomputed minimal irreducible invariant basis for each rank-k tensor under SO(3) and the crystal space group, and the network is architecturally constrained to output only the scalar coefficients in this basis. By construction, any output of this head satisfies all index and symmetry constraints exactly, so the architectural design of the head is directly driven by the requirement of strict symmetry for high-order tensors.
> 2. **Unified head for rank-2/3/4 tensors, decoupled from ad-hoc schemes**. The same irreducible-basis head is used for dielectric (rank-2), piezoelectric (rank-3), and elastic (rank-4) tensors, without redesigning tensor-specific perturbations or response modules. Architecturally, this gives a single, plug-and-play, symmetry-guaranteed tensor head that can be attached to any O(3)-equivariant backbone (here, an SO(2)-reduced eSCN). This is different from prior architectures (e.g., GMTNet), which operate in the full component or perturbation space and rely on post-hoc symmetry enforcement that can still leave small violations.
> 3. **Backbone–head integration tailored to high-order symmetry constraints**. While the convolution blocks themselves follow standard spherical-harmonic designs, we adapt and integrate them with the irreducible-basis head (SO(2)-reduced O(3) convolutions, gated irreps blocks, depth-wise channel expansion) so that the backbone produces rich equivariant features that are efficiently mapped into the symmetry-reduced coefficient space in a single forward pass. This combination yields an architecture that is both practical for high-order tensors and, by design, strictly respects all required symmetries.
>
> We hope this clarifies the reviewer’s question.

---

> ### Author Response · Authors · 2025-11-21
>
> **Question 4.** Please provide a complexity analysis (runtime and memory) for the basis generation step in Algorithm 1.
>
> **Answer 4.** We thank the reviewer's suggestions.
>
> a. **Complexity of basis generation.** Algorithm 1 consists of two stages: (i) the unconstrained $SO(3)$ irreducible decomposition via GenerateSO3Irreps(k) and (ii) the symmetry-compatibility filtering ApplySym($M_0$, $G_{\text{sym}}$), with an optional projector step to obtain explicit invariant bases. The function GenerateSO3Irreps(k) enumerates all partitions of the tensor rank $k$ with at most 3 parts and applies the $SU(3)\to SO(3)$ branching rules (Appendix F). The number of such partitions grows as $p_3(k) = \Theta(k^2)$, and each branching call has quadratic cost in $k$, leading to an overall runtime of $O(k^4)$ and $O(|L_k|) = O(k)$ memory for storing the map $M_0[\ell]$, where $L_k$ is the set of distinct spins. The symmetry step ApplySym($M_0$, $G_{\text{sym}}$) loops over all $\ell \in L_k$ and, for each spin, evaluates the $O(3)$ character on all conjugacy classes of the point group $G_{\text{sym}}$. Its runtime is therefore $O(|L_k|\cdot C_G) = O(k\cdot C_G)$, where $C_G$ is the number of conjugacy classes (a small constant for crystallographic point groups), with again $O(k)$ memory. When an explicit invariant basis is required, we form the projectors $\Pi_\ell = |G_{\text{sym}}|^{-1} \sum_{g \in G_{\text{sym}}} \rho^{(\ell)}(g).$
>
> on each $SO(3)$ block, which costs $O\bigl(|G_{\text{sym}}| d_\ell^2 + d_\ell^3\bigr)$ per spin, where $d_\ell = 2\ell + 1$. Summed over all $\ell \le k$, this yields at most $O\bigl(|G_{\text{sym}}| k^3 + k^4\bigr)$ time and $O(k^3)$ memory. For the tensor orders considered in this work ($k \le 4$), the entire basis generation is a one-off precomputation that takes negligible time and memory compared to training or inference of the neural network.
>
> b. **Case Study.** For the elastic tensor (rank-4), Algorithm 1 first performs an unconstrained $SO(3)$ decomposition of $V^{\otimes 4}$ and then imposes the major/minor index symmetries of the elastic stiffness tensor. This reduces the 81-dimensional space of rank-4 tensors to the 21-dimensional elasticity space, which decomposes under $SO(3)$ as $\mathcal{C} \cong 2 \times \ell = 0 ;\oplus; 2 \times \ell = 2 ;\oplus; 1 \times \ell = 4$, i.e. $M_0^{\text{elastic}}[0] = 2$, $M_0^{\text{elastic}}[2] = 2$, and $M_0^{\text{elastic}}[4] = 1$. The corresponding dimensions $2 \times (2\cdot 0 + 1) + 2 \times (2\cdot 2 + 1) + 1 \times (2\cdot 4 + 1) = 2 + 10 + 9 = 21$ match the 21 independent components of the elastic tensor. Given a crystal point group $G_{\text{sym}}$, the procedure ApplySym then computes, for each spin $\ell$, the multiplicity $n_\ell$ of the trivial representation in the restriction of the $(2\ell+1)$-dimensional $SO(3)$ irrep to $G_{\text{sym}}$ via a class-sum character test. For a cubic crystal with $G_{\text{sym}} = O_h$, this yields $n_0 = 1$, $n_2 = 0$, $n_4 = 1$, and hence $M_{\text{final}}^{\text{elastic}}[0] = 2 \times n_0 = 2$, $M_{\text{final}}^{\text{elastic}}[2] = 2 \times n_2 = 0$, $M_{\text{final}}^{\text{elastic}}[4] = 1 \times n_4 = 1$. The total number of invariant coefficients is therefore $2 + 0 + 1 = 3$, in one-to-one correspondence with the three independent elastic constants $(C_{11}, C_{12}, C_{44})$ of a cubic crystal. For each spin $\ell$, we finally construct explicit invariant bases $\{\mathcal{B} _ a^{(\ell)}\} _ {a=1}^{M_{\mathrm{final}}^{\mathrm{elastic}}[\ell]}$ via the projector $\Pi_\ell = |G_{\text{sym}}|^{-1} \sum_{g \in G_{\text{sym}}} \rho^{(\ell)}(g)$. The network predicts only the scalar coefficients of these irreducible basis tensors, so any reconstructed elastic tensor is guaranteed to satisfy both the index symmetries and the crystal point-group symmetry by construction.
>
>
> c.   **Exact runtime and memory usage.** Practical cost of basis construction. We also measured the actual runtime and memory usage of our PyTorch+e3nn implementation on the same machine used for all experiments, equipped with four NVIDIA RTX 3090 GPUs, an Intel Xeon Gold 5120 CPU @ 2.20 GHz, 512 GB of system memory, and running Ubuntu 22.04. For a batch of 128 tensors, constructing the unconstrained rank-2/3/4 Cartesian tensor decompositions takes 0.20 s / 0.32 s / 0.55 s with additional memory of at most 8 MB, and applying the corresponding forward and inverse transforms costs ≈0.1–0.6 s with only a few MB of extra memory. Adding the index-symmetry constraints increases the one-time construction cost to 0.29 s (rank-2), 0.94 s (rank-3), and 1.31 s (rank-4), with at most ~8 MB additional memory, while the symmetry-aware forward transform for rank-4 takes 0.53 s and ~10.5 MB for a batch of 128 tensors. These numbers confirm that basis generation and symmetry-constrained tensor/irreps transforms are lightweight one-off (or per-batch) operations compared to overall network training.

---

> ### Author Response · Authors · 2025-11-21
>
> **Question 5** Since the model predicts scalar coefficients, do these coefficients correlate with any known physical or geometric scalar invariants of the tensor (e.g., the bulk modulus in elasticity)? Exploring this connection could enhance the interpretability of the model’s predictions.
>
> **Answer 5.** We thank the reviewer for this question. Our model always predicts scalar coefficients associated with a symmetry-constrained invariant basis. By construction, each coefficient is invariant under global rotations and the crystal point group. For a fixed tensor type, these coefficients span the same invariant subspace as the standard physical scalar quantities used in that domain; they should therefore be viewed as coordinates in the space of symmetry-allowed scalar invariants.
>
> For **dielectric tensors** (rank-2, symmetric), the invariant basis naturally separates an isotropic component (trace-like, representing an average permittivity) and a small number of anisotropic components that quantify deviations between principal dielectric constants [1]. Our predicted coefficients are linear combinations of these objects, and hence are linearly related to familiar quantities such as the average dielectric constant and measures of dielectric anisotropy for each crystal class.
>
> For **piezoelectric tensors** (rank-3, with intrinsic index symmetry), the invariant basis encodes symmetry-distinct coupling channels between polarization and strain (e.g., longitudinal vs transverse vs shear-type couplings). In any given crystal system, the resulting coefficients span the same invariant subspace as the conventional piezoelectric constants (such as $d_{33}, d_{31}, d_{15}$), so there exists a fixed linear map between our coefficients and any chosen set of standard piezoelectric moduli [2].
>
> For **elastic tensors** (rank-4, with major/minor symmetry), the invariant decomposition recovers the classical split into scalar, quadrupolar, and higher-order anisotropic parts. After restricting to the crystal point group, the remaining coefficients span the same low-dimensional invariant space as the usual elastic constants; for example, in cubic symmetry the three predicted coefficients span the same 3-dimensional subspace as ($C_{11}, C_{12}, C_{44}$) and their associated bulk/shear-type moduli and anisotropy measures [3,4].
>
> In this work we use an orthonormal invariant basis obtained via group-theoretic projectors, rather than explicitly aligning each basis element with a named physical modulus; standard physical scalars are therefore given by fixed linear combinations of our coefficients. A tensor-specific, more detailed discussion (including explicit examples for dielectric, piezoelectric, and elastic tensors in symmetry constraints) is provided in the Appendix.
>
>
>
> [1] Kocot, Antoni, et al. "Dielectric study of liquid crystal dimers: probing the orientational order and molecular interactions in nematic and twist-Bend nematic Phases." The Journal of Physical Chemistry B 127.31 (2023): 7082-7090.
>
> [2] Gorfman, Semën, and Nan Zhang. "Piezoelectric Coefficients and Crystallographic Symmetry." *Piezoelectric Materials: From Fundamentals to Emerging Applications* 1 (2024): 1-15.
>  [3] Browaeys, Jules Thomas, and Sébastien Chevrot. "Decomposition of the elastic tensor and geophysical applications." *Geophysical Journal International* 159.2 (2004): 667-678.
>
> [4] Auffray, Nicolas. "On the isotropic moduli of 2D strain-gradient elasticity." *Continuum Mechanics and Thermodynamics* 27.1 (2015): 5-19.

---

### Official Review · Reviewer_o2vR · 2025-11-10

**Soundness:** 3
**Presentation:** 1
**Contribution:** 2
**Rating:** 4
**Confidence:** 4

**Summary:**

The paper predicts symmetry-constrained SO(3) irreducible components and reconstructs crystal tensors to guarantee exact symmetry; IrredNet implements this, outperforming MEGNet/ETGNN/GMTNet on JARVIS-DFT (especially elastic) and improving downstream moduli, with ablations confirming irreducible-space prediction plus explicit symmetry are essential.

**Strengths:**

1. IrredNet predicts SO(3) irreducible components and reconstructs tensors only from symmetry-compatible bases, guaranteeing exact crystal/index symmetries without post-hoc fixing.
2. IrredNet shows the gains specifically come from irreducible-space prediction and explicit symmetry constraints.

**Weaknesses:**

### Majors

1. **Contributions of this work is incremental**. Computing higher-order tensor decompositions and invariant bases under SO(3)/crystal symmetries is routine and well-known; the paper mainly engineers these known tools into an ML pipeline to guarantee symmetry rather than introducing new theory.
2. **Presentation can be improved**. It is recommended to add a concrete running example (e.g., rank-4 elastic) walking through your algorithm and include a brief contrast explaining why prior methods (e.g., GMTNet/ETGNN) can still violate high-order symmetries without explicit projection. Also, please redo tables with self-contained, consistent captions including dataset, property, baselines, and metrics+units.

### Minors

1. Line 314, 317, 776 use "nonzero" but line 412, 759, 950 use "non-zero". Please make them consistent.
2. Line 105 "Jarvis" -> "JARVIS"
3. Line 401: "GMTNET" -> "GMTNet"

**Questions:**

1. In what precise way does predicting SO(3) irreducible components differ from prior equivariant GNNs (like GMTNet) that already operate in irreps?
2. How sensitive are results to $\ell_{max}$ and channel counts?

---

> ### Author Response · Authors · 2025-11-21
>
> We thank the reviewer for the careful reading, constructive suggestions, and detailed questions. Below, we address the reviewer’s concerns and questions accordingly.
>
> **Question 1. Contributions of this work is incremental**. Computing higher-order tensor decompositions and invariant bases under SO(3)/crystal symmetries is routine and well-known; the paper mainly engineers these known tools into an ML pipeline to guarantee symmetry rather than introducing new theory.
>
> **Answer 1.** We thank the reviewer’s valuable comment. Our contribution is to turn these tools into a unified, practically effective framework for high-order crystal tensor prediction. Concretely, this work goes beyond a routine engineering of known formulas in the following ways:
>
> 1. **New output parameterization, not just equivariant features**. Prior crystal GNNs (MEGNet, ETGNN, GMTNet) either treat tensor components as (almost) independent scalars with post-hoc symmetrization, or operate in the full component space with perturbation-based reconstruction and learned projection blocks. In contrast, we explicitly construct a minimal irreducible invariant basis for each rank-k tensor under symmetry constraints, and parameterize the model only in terms of the scalar coefficients in this basis. To our knowledge, such a strict symmetry-complete output parameterization, shared across rank-2/3/4 crystal tensors, has not been used in prior ML tensor predictors.
> 2. **Unified, scalable framework for realistic high-order tensors**. Making these decompositions usable for real crystals is nontrivial: we handle up to fourth-order tensors under full symmetry and implement a single irreducible-basis head that works for dielectric, piezoelectric, and elastic tensors without redesigning perturbations or tensor-specific heads. This head can in principle be attached to any SO(3)-equivariant backbone (here, an efficient SO(2)-reduced eSCN), yielding a general plug-and-play module rather than a task-specific engineering trick.
>
> 3. **Strict guarantees and nontrivial empirical gains**. Because we only predict coefficients in the symmetry-reduced basis, all index and symmetry constraints are satisfied exactly by construction, which is stronger than the approximate, post-hoc enforcement in prior work. At the same time, on the same benchmarks and comparable budgets, our model consistently outperforms strong equivariant baselines (including GMTNet) on all three tensor tasks. These systematic accuracy gains, together with strict guarantees, indicate that the proposed formulation is more than a minor incremental tweak.
>
> We hope this clarifies the reviewer’s question.

---

> ### Author Response · Authors · 2025-11-21
>
> **Question 2. Presentation can be improved**. It is recommended to add a concrete running example (e.g., rank-4 elastic) walking through your algorithm and include a brief contrast explaining why prior methods (e.g., GMTNet/ETGNN) can still violate high-order symmetries without explicit projection. Also, please redo tables with self-contained, consistent captions including dataset, property, baselines, and metrics+units.
>
> **Answer 2.** We thank the reviewer for these suggestions on improving the presentation, and we provide a detailed description of the exact computational procedure in the Appendix.
>
> (1) We explicitly decompose Algorithm 1 into two stages: (i) an unconstrained $SO(3)$ decomposition via GenerateSO3Irreps(k), which enumerates all partitions of the tensor rank $k$ with at most three parts and applies the $SU(3)\to SO(3)$ branching rules, and (ii) a symmetry-compatibility filtering step ApplySym($M_0$, $G_{\text{sym}}$) that incorporates index and point-group symmetries. We show that GenerateSO3Irreps(k) runs in $O(k^4)$ time with $O(k)$ memory (for the list of $SO(3)$ spins), while ApplySym runs in $O(k\cdot C_G)$ time and $O(k)$ memory, where $C_G$ is the number of conjugacy classes of the crystal point group. When an explicit invariant basis is needed, we form projectors  $\Pi_\ell$  on each spin, which in total costs at most $O\bigl(|G_{\text{sym}}|,k^3 + k^4\bigr)$ time and $O(k^3)$ memory. For the tensor orders considered in this work ($k \le 4$), we emphasize that this basis generation is a one-off precomputation with negligible cost compared to network training or inference. (2) We add a concrete running example for the rank-4 elastic tensor. We show how Algorithm 1 reduces the 81-dimensional space of rank-4 tensors to the 21-dimensional elasticity space with $SO(3)$ decomposition $\mathcal{C} \cong 2 \times \ell = 0 \;\oplus\; 2 \times \ell = 2 \;\oplus\; 1 \times \ell = 4$, and then, for a cubic crystal with $G_{\text{sym}} = O_h$, further reduces this to just three invariant coefficients corresponding exactly to $C_{11}, C_{12}, C_{44}$. We also explain how the projectors $\Pi_\ell$ are used to construct explicit invariant basis tensors ${B_a^{(\ell)}}$, and that IrredNet predicts only the scalar coefficients of these basis tensors. This concrete example makes it clear how the algorithm works in practice and why any reconstructed elastic tensor necessarily satisfies both index symmetries and crystal point-group symmetry by construction.
>
> Second, we have included a brief contrast with GMTNet and ETGNN in the appendix. Specifically, we explain how GMTNet uses an equivariant backbone plus perturbation-based energy responses and a learned projection in the full component space, and how ETGNN constructs tensor outputs from edge outer products; in both cases, symmetries are encouraged but not guaranteed at the level of the final components, so small violations of zero-elements/equalities can persist in practice if no explicit projection onto the fully symmetry-reduced space is applied. In contrast, because IrredNet operates directly in the irreducible symmetry-complete basis, any predicted coefficient vector automatically yields a tensor that satisfies all index and space-group symmetries by construction.
>
> Finally, we have  revised the tables to make them fully self-contained and consistent. The revised captions clearly state, for each table, the dataset (e.g., JARVIS-DFT), the predicted property (dielectric, piezoelectric, or elastic tensor), the list of baselines (MEGNet, ETGNN, GMTNet, etc.), and the evaluation metrics including units (e.g., Fnorm in the corresponding tensor units, EwT as a percentage, and wall-clock time in seconds). This will make it easier for readers to interpret the results without repeatedly referring back to the main text.
>
> We hope this answer addresses the reviewer’s comment.

---

> ### Author Response · Authors · 2025-11-21
>
> **Question 3.** In what precise way does predicting SO(3) irreducible components differ from prior equivariant GNNs (like GMTNet) that already operate in irreps?
>
> **Answer 3.** We appreciate the reviewer’s question. The key difference lies in what the model predicts and how the prediction relates to the underlying tensor property.
>
> •  In our method, the final predicted quantities are directly the SO(3) irreducible components of the target tensor property. In other words, our output irreps correspond exactly to the mathematical decomposition of the physical tensor into its SO(3) irreducible parts. This creates a one-to-one correspondence between the model’s prediction and the true tensor decomposition.
>
> •  In previous methods like GMTNet, although the internal feature representations are also organized into irreps, the predicted outputs are not the irreducible components of the tensor property itself. They predict the property through an indirect perturbation–response mechanism, rather than by explicitly predicting the irreducible pieces of the tensor. Thus, its use of irreps is architectural but not tied to the tensor property’s exact SO(3) decomposition.
>
> So the novelty is not merely “using irreps,” which is common, but predicting the irreps of the physical property itself, yielding a prediction space that aligns exactly with the theoretical decomposition of the target tensor. We hope this answer addresses the reviewer’s comment.

---

> ### Author Response · Authors · 2025-11-21
>
> **Question 4.** How sensitive are results to $\ell_{\max}$ and channels?
>
> **Answer 4-1.** We thank the reviewer for raising this question. Our choice of $\ell_{\max}$ is driven by both representation theory and comparability with prior work. From $SO(3)$ representation theory, the target tensor only needs irreps up to a certain order: symmetric rank-2 tensors (dielectric) require $\ell \le 2$, pair-symmetric rank-3 tensors (piezoelectric) require $\ell \le 3$, and symmetric rank-4 tensors (elastic) require $\ell \le 4$. In principle, a model predicting only a symmetric rank-2 tensor could already be implemented with $\ell_{\max} = 2$. In our experiments, we choose $\ell_{\max} = 3$ for dielectric and piezoelectric tensors and $\ell_{\max} = 4$ for elastic tensors: this is slightly more conservative than the minimal requirement, so the backbone can use intermediate angular features while the final tensor head only uses irreps consistent with the target index symmetries. These settings are also consistent with the hyperparameter ranges reported for GMTNet, which facilitates a fair comparison.
>
> To verify that our results are not overly sensitive to this choice, we perform an explicit $\ell_{\max}$ ablation (Table 5). For dielectric tensors, $\ell_{\max} = 2$ vs. $3$ yields almost identical accuracy ($F_{\text{norm}}$ 3.31 vs. 3.23; EwT@25% 85.9% vs. 85.8%), with only a modest increase in wall-clock time (711 s vs. 798 s). For piezoelectric tensors, $\ell_{\max} = 3,4,5$ all give very similar performance ($F_{\text{norm}}$ 0.29–0.33, EwT@25% around 49%), while runtime increases from 288 s to 454 s. For elastic tensors, $\ell_{\max} = 4,5,6$ show nearly identical EwT@25% (66.5–66.7%), but the total time grows from 14121 s to 18711 s. Overall, once the physically required degrees are included, increasing $\ell_{\max}$ mainly increases computational cost and does not systematically improve accuracy, indicating that our conclusions are robust to the precise choice of $\ell_{\max}$.
>
> | Dataset       | ℓ_max | Fnorm (↓) | EwT 25% (↑) | EwT 10% (↑) | EwT 5% (↑) | Total Time (s) (↓) |
> | ------------- | :---: | --------: | ----------: | ----------: | ---------: | -----------------: |
> | Dielectric    |   2   |      3.31 |   **85.9%** |       52.1% |  **29.9%** |            **711** |
> | Dielectric    |   3   |  **3.23** |       85.8% |   **53.2%** |      29.8% |                798 |
> | Piezoelectric |   3   |  **0.29** |       49.3% |       49.3% |  **49.2%** |            **288** |
> | Piezoelectric |   4   |      0.30 |       49.1% |   **48.9%** |      48.5% |                314 |
> | Piezoelectric |   5   |      0.33 |       49.2% |   **48.9%** |      48.6% |                454 |
> | Elastic       |   4   |     64.23 |   **66.5%** |       26.5% |  **12.1%** |          **14121** |
> | Elastic       |   5   |     64.61 |       65.7% |       26.1% |  **12.1%** |              16237 |
> | Elastic       |   6   | **63.88** |   **66.5%** |   **26.9%** |      12.0% |              18711 |

---

> > ### Author Response · Authors · 2025-11-21
> >
> > **Answer 4-2.** Regarding the channel sensitivity, we allocate more channels to low and mid degrees ($\ell = 0,1,2$), which carry most of the physical signal, and fewer channels to higher degrees. This strategy provides sufficient capacity to model complex high-order anisotropy while maintaining stable training and reasonable runtime. For each degree $\ell$ and parity (even/odd), we assign channels in a monotonically decreasing manner with $\ell$, roughly following a geometric decay (e.g., $32,16,8,4,2$ for $\ell = 0,1,2,3,4$). Low and mid degrees ($\ell = 0,1,2$) are given significantly more channels because they carry most of the physical signal (scalar, vector, and low-order anisotropy), while higher degrees ($\ell \ge 3$) are assigned fewer channels and mainly serve to capture fine-grained anisotropy. Even and odd irreps at the same $\ell$ share the same width, reflecting that we do not impose any prior bias between parities. This degree-wise allocation gives a good trade-off between expressivity of high-order components and computational efficiency.
> >
> >
> > To assess the sensitivity of this design, we perform a channel-allocation ablation under a comparable total width (Table~6). Besides our default monotonically decreasing allocation (e.g., $32\times 0_e/0_o$, $16\times 1_e/1_o$, $8\times 2_e/2_o$, $4\times 3_e/3_o$, $2\times 4_e/4_o$), we consider two alternatives: (i) an approximately flat allocation with 12 channels per parity at each degree (e.g., $12\times 0_e/0_o, \dots, 12\times 4_e/4_o$), where the value 12 is chosen so that the overall number of channels is similar to the default setting, and (ii) a high-degree-heavy allocation that reverses the pattern and allocates most channels to $\ell = 3,4$. Across all three tensor prediction tasks, our default scheme achieves the best or near-best $F_{\text{norm}}$ and $\mathrm{EwT}$, while the high-degree-heavy configuration clearly degrades accuracy (e.g., for elastic tensors, $F_{\text{norm}}$ increases from $64.23$ to $69.77$ and $\mathrm{EwT@25\%}$ drops from $66.5\%$ to $61.5\%$). The flat allocation is only slightly worse than the default, indicating that the model is reasonably robust to moderate changes in the exact per-degree counts. Overall, these results support our design principle: allocating more capacity to low and mid degrees ($\ell = 0,1,2$) is important for performance, whereas over-emphasizing high degrees is not beneficial and mainly increases computational cost. We hope this response addresses the reviewer’s concerns.
> >
> > | Dataset       | hidden irreps                                               | Fnorm (↓) | EwT 25% (↑) | EwT 10% (↑) | EwT 5% (↑) |
> > | ------------- | ----------------------------------------------------------- | --------- | ----------- | ----------- | ---------- |
> > | Dielectric    | 32x0e+32x0o+16x1e+16x1o+8x2e+8x2o+4x3e+4x3o                 | **3.23**  | **85.8%**   | **53.2%**   | **29.8%**  |
> > | Dielectric    | 12x0e+12x0o+12x1e+12x1o+12x2e+12x2o+12x3e+12x3o             | 3.43      | 83.8%       | 51.2%       | 26.7%      |
> > | Dielectric    | 12x0e+12x0o+12x1e+12x1o+12x2e+12x2o+12x3e+12x3o             | 3.86      | 78.3%       | 40.9%       | 20.4%      |
> > | Piezoelectric | 32x0e+32x0o+16x1e+16x1o+8x2e+8x2o+4x3e+4x3o                 | **0.29**  | **49.3%**   | **49.3%**   | **49.2%**  |
> > | Piezoelectric | 12x0e+12x0o+12x1e+12x1o+12x2e+12x2o+12x3e+12x3o             | 0.33      | 48.9%       | 48.7%       | 48.4%      |
> > | Piezoelectric | 12x0e+12x0o+12x1e+12x1o+12x2e+12x2o+12x3e+12x3o             | 0.41      | 47.3%       | 47.2%       | 46.8%      |
> > | Elastic       | 32x0e+32x0o+16x1e+16x1o+8x2e+8x2o+4x3e+4x3o+2x4e+2x4o       | **64.23** | 66.5%       | **26.5%**   | **12.1%**  |
> > | Elastic       | 12x0e+12x0o+12x1e+12x1o+12x2e+12x2o+12x3e+12x3o+12x4e+12x4o | 64.41     | **66.8%**   | 23.4%       | 11.5%      |
> > | Elastic       | 2x0e+2x0o+4x1e+4x1o+8x2e+8x2o+16x3e+16x3o+32x4e+32x4o       | 69.77     | 61.5%       | 20.9%       | 7.5%       |

---

### Meta-Review · Area_Chair_ocnw · 2026-01-04

**Summary:**

Reviewers raised concerns regarding the novelty of the contribution, the lack of clarity and motivation in the empirical evaluation, and the overall presentation of the paper. Although several aspects have been addressed, crucial concerns, the novelty, motivation, and presentation, remain unresolved. I have carefully read the paper and agree with these assessments. I therefore recommend rejection of the paper.

**Reviewer Concerns:**

The major concern raised by several reviewers, the novelty of the contribution, remains outstanding. There is also room for improvement in the presentation of the paper, particularly through a more careful discussion of related work.

In contrast, concerns regarding the details of the empirical evaluation, such as runtime comparisons and the complexity analysis of the proposed method, have been addressed in the revised manuscript and the authors’ rebuttal.

**Reviewer Scores:**

**Reviewer o2vR** raised concerns about the novelty of the contribution, the presentation of the paper, and several unclear aspects of the experimental results. Although the authors clarified the motivation in the rebuttal, the response does not sufficiently address these concerns, particularly regarding novelty. Therefore, this reviewer is likely to maintain the score.

**Reviewer qDLZ** raised concerns about the empirical evaluation, the overall clarity of the paper, and its novelty. While the rebuttal partially addresses some issues, such as runtime evaluation, the concern regarding novelty remains unresolved. Thus, this reviewer is likely to maintain the score.

**Reviewer Qp15** raised concerns about the presentation of experimental results, inconsistencies between the stated motivation and the evaluation, and the experimental design. Although the authors attempted to address these issues, it is fundamentally difficult to fully resolve such critical concerns at this stage. As a result, while there is a possibility that the reviewer may increase the score, it is likely to remain on the negative side.

**Reviewer eYBb** pointed out issues related to missing discussions and citations. Although the authors provided responses, the reviewer has already replied and confirmed that these concerns have not been resolved. Therefore, this reviewer is likely to maintain the score.

---

### Decision · Program_Chairs · 2026-01-26

Reject